# Native architecture of the *Chlamydomonas* chloroplast revealed by in situ cryo-electron tomography

Benjamin D Engel*[†], Miroslava Schaffer[†], Luis Kuhn Cuellar, Elizabeth Villa[‡], Jürgen M Plitzko[§], Wolfgang Baumeister*

Department of Molecular Structural Biology, Max Planck Institute of Biochemistry, Martinsried, Germany

**\*For correspondence:**
engelben@biochem.mpg.de
(BDE); baumeist@biochem.
mpg.de (WB)

[†]These authors contributed equally to this work

**Present address:** [‡]Department of Chemistry and Biochemistry, University of California, San Diego, La Jolla, United States; [§]Department of Chemistry, Bijvoet Center for Biomolecular Research, Utrecht University, Utrecht, Netherlands

**Competing interests:** The authors declare that no competing interests exist.

**Reviewing editor**: Joanne Chory, Salk Institute, United States

**Abstract** Chloroplast function is orchestrated by the organelle's intricate architecture. By combining cryo-focused ion beam milling of vitreous *Chlamydomonas* cells with cryo-electron tomography, we acquired three-dimensional structures of the chloroplast in its native state within the cell. Chloroplast envelope inner membrane invaginations were frequently found in close association with thylakoid tips, and the tips of multiple thylakoid stacks converged at dynamic sites on the chloroplast envelope, implicating lipid transport in thylakoid biogenesis. Subtomogram averaging and nearest neighbor analysis revealed that RuBisCO complexes were hexagonally packed within the pyrenoid, with ~15 nm between their centers. Thylakoid stacks and the pyrenoid were connected by cylindrical pyrenoid tubules, physically bridging the sites of light-dependent photosynthesis and light-independent carbon fixation. Multiple parallel minitubules were bundled within each pyrenoid tubule, possibly serving as conduits for the targeted one-dimensional diffusion of small molecules such as ATP and sugars between the chloroplast stroma and the pyrenoid matrix.

## Introduction

The chloroplast is the site of eukaryotic photosynthesis and carbon fixation, coupled processes that convert the energy of light into stored biochemical energy, while exchanging environmental $CO_2$ for $O_2$. As such, the chloroplast both directly and indirectly sustains much of the life on our planet.

The biochemical pathways of photosynthesis rely on the chloroplast's elaborate architecture. In both higher plants and algae, the light-dependent photosynthetic reactions take place in the thylakoids, sheet-like membrane-bound compartments that group together into regularly-spaced stacks (called grana in higher plants). Photons are absorbed by photosystem complexes within the thylakoid membrane, and the downstream electron transfer chain generates NADPH in the chloroplast stroma. This process, combined with the splitting of $H_2O$ into $O_2$ and $H^+$, concentrates $H^+$ inside the thylakoid lumen. The resulting proton gradient across the thylakoid membrane is used by the ATP synthase complex to generate ATP in the chloroplast stroma, where the light-independent carbon fixation reactions occur. ATP and NADPH power the Calvin–Benson cycle, a series of reactions that incorporate $CO_2$ into the three-carbon sugar glyceraldehyde 3-phosphate (G3P). Downstream pathways produce starch, which is stored within the chloroplast. The $CO_2$-fixing step of the Calvin–Benson cycle is catalyzed by the octomeric ribulose-1,5-bisphosphate carboxylase/oxygenase (RuBisCO) complex. In higher plants, which have ready access to gaseous $CO_2$, RuBisCO complexes are distributed throughout the chloroplast stroma. However, algae primarily live in aquatic environments where $CO_2$ is dissolved as $HCO_3^-$. In order to maintain efficient carbon fixation, the majority of algal RuBisCO complexes are concentrated in a specialized region of the chloroplast known as the pyrenoid (*Holdsworth, 1971*; *Lacoste-Royal and Gibbs, 1987*; *Borkhsenious et al., 1998*), where carbonic anhydrase frees $CO_2$

**eLife digest** Many organisms can harvest light to produce their own energy through a process called photosynthesis. In plant and algal cells, photosynthesis takes place within the chloroplasts, which are compartments that contain stacks of structures called thylakoids.

Inside the thylakoids, proteins absorb energy from light and convert it into biochemical energy that can be used by the cell. This energy then powers a series of reactions that result in carbon dioxide being incorporated into energy-rich sugars. The enzyme RuBisCO is essential for this process, and is believed to be the most abundant protein on Earth. In land plants, RuBisCO is found throughout the chloroplast, but in algae it is limited to a specialized area called the pyrenoid.

Much of our current knowledge of chloroplast structure comes from transmission electron microscopy (TEM) images. However, the traditional methods used to prepare cells for TEM can damage their internal structures. Also, previous studies have focused primarily on the chloroplasts of land plants, even though aquatic organisms—including the alga *Chlamydomonas*—account for over 50% of photosynthesis on the planet.

Here, Engel et al. provide the first three-dimensional structures of *Chlamydomonas* chloroplasts in their natural state. They used several recently-developed techniques to study cells that were preserved in a close-to-living condition. The cells were rapidly frozen, thinned with a technique called cryo-focused ion beam milling, and then imaged by a type of TEM called cryo-electron tomography.

The three-dimensional images provide many insights into the *Chlamydomonas* chloroplast, including evidence that lipids and proteins move between the membrane that surrounds the chloroplast—called the chloroplast envelope—and the tips of the thylakoids. These images show how thylakoids may be built by the transport of molecules from the chloroplast envelope. In addition, the images reveal the detailed structures of the tubes that connect the thylakoids to the pyrenoid, which could explain how the two stages of photosynthesis (light harvesting and the conversion of carbon dioxide) can be coordinated even though they occur at different places within the chloroplast.

Engel et al. also observed that RuBisCO enzymes are arranged in a hexagonal pattern inside the pyrenoid, but are spaced too far apart to make direct contact with each other. To understand how the pyrenoid is assembled, a future goal will be to determine what causes RuBisCO to be arranged in this way.

from solution (*Karlsson et al., 1998*; *Moroney et al., 2011*). Thus, in algae the light-dependent and light-independent reactions of photosynthesis are physically segregated to distinct regions of the chloroplast (*Figure 1A*). While the chloroplast's biochemical pathways are well characterized, a better understanding of the organelle's native three-dimensional (3D) architecture would help answer questions about the biogenesis of thylakoids and pyrenoids, as well as the cellular mechanisms that link the activities of these two physically separated systems.

Much of our understanding of chloroplast architecture comes from conventional transmission electron microscopy (TEM) studies (*Hodge et al., 1955*; *Sager and Palade, 1957*; *Heslop-Harrison, 1963*; *Ohad et al., 1967a*, *1967b*) using the traditional protocol of sample fixation, heavy metal staining, dehydration, plastic embedding, and sectioning with an ultramicrotome. Each of these preparation steps can distort structures and obscure high-resolution information (*Weston et al., 2009*). Sample preparation by freeze-fracture has enabled observations of integral proteins embedded in membrane surfaces (*Park and Pfeifhofer, 1969*; *Goodenough and Staehelin, 1971*), but it cannot provide a 3D view of chloroplast architecture. Vitrification by rapid freezing offers the best possible preservation of biological material (*Dubochet et al., 1988*). Imaging vitreous samples by cryo-electron tomography (cryo-ET) generates 3D views of native structures, where image contrast corresponds to the intrinsic variation in mass density instead of the sample's local affinity for heavy metal stains (*Baumeister et al., 1999*).

Over the last decade, there has been progress in the 3D visualization of photosynthetic structures. Much of this work has focused on cyanobacteria (*Nevo et al., 2007*; *Ting et al., 2007*; *Konorty et al., 2008*, *2009*; *Iancu et al., 2010*; *Liberton et al., 2011a*, *2011b*), as the small size of these prokaryotes allows imaging by whole-cell tomography. With the exception of a few very small cells

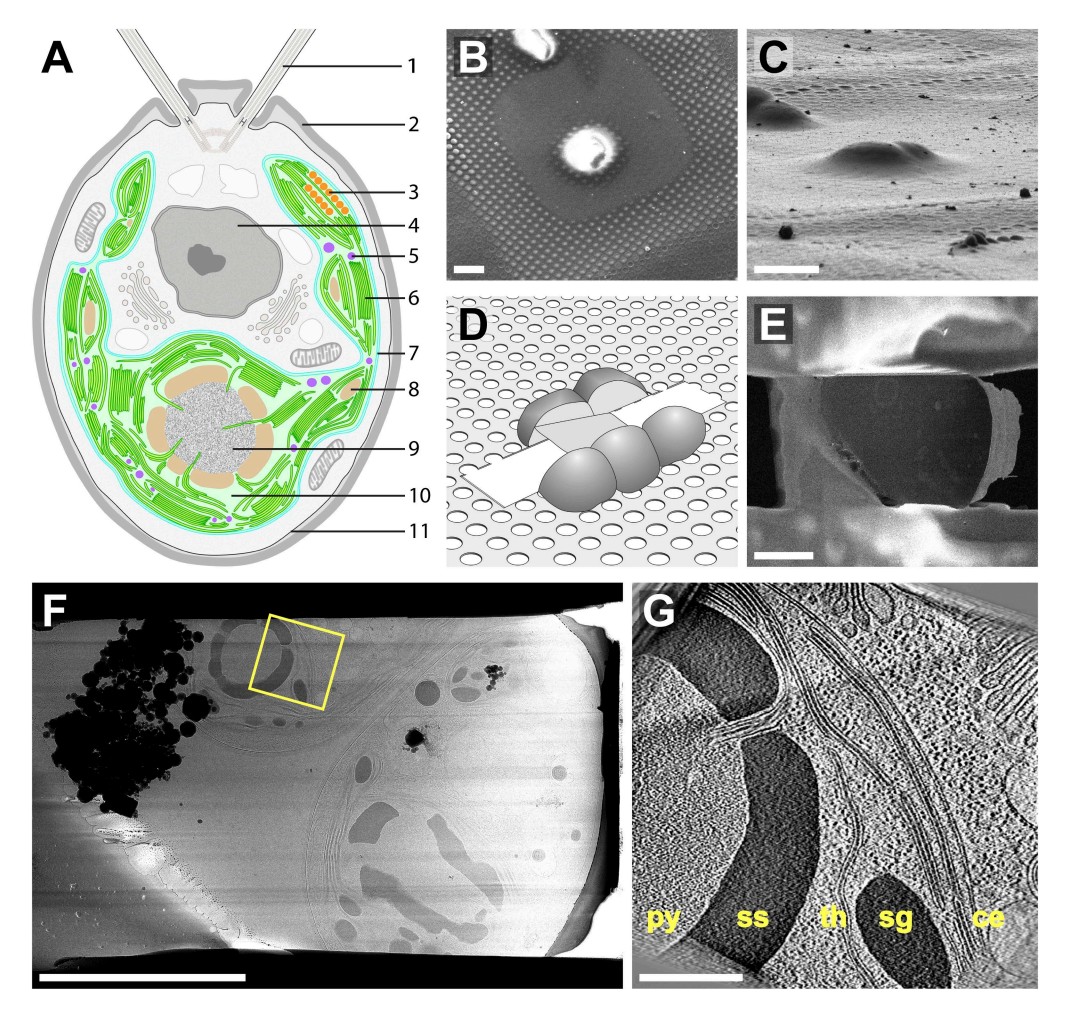

**Figure 1**. Cryo-FIB of plunge-frozen *Chlamydomonas* cells. (**A**) Cross-section diagram of a *Chlamydomonas* cell. The chloroplast is shown in color. Labels: (1) flagellum, (2) cell wall, (3) eyespot, (4) nucleus, (5) plastoglobule, (6) thylakoids, (7) chloroplast envelope, (8) starch granule, (9) pyrenoid, (10) chloroplast stroma, (11) plasma membrane. (**B**–**E**) Preparation of FIB lamellas in the dual-beam microscope. (**B**) Top-view scanning electron microscopy (SEM) image of *Chlamydomonas* cells frozen onto the holey carbon of a 200 mesh grid. (**C**) The same cells imaged with the FIB (secondary electron detection) at the milling angle of 9°. (**D**) Schematic of a finished lamella. The direction of FIB milling is from left to right. (**E**) SEM top-view of the lamella milled from the cells shown in **B** and **C**. The direction of FIB milling is from right to left. (**F**) TEM high-defocus montaged overview of the same lamella. Unbinned pixel size: 30.1 Å. (**G**) Slice from a tomographic volume acquired at the position indicated by the yellow box in **F**. From left to right, RuBisCO complexes of the pyrenoid (py) are surrounded by a starch sheath (ss), followed by thylakoids (th), a starch granule (sg), and the chloroplast envelope (ce). Outside of the chloroplast, Golgi stacks and rough endoplasmic reticulum can be seen. Both cytoplasmic and chloroplast ribosomes are also visible. The tomogram was 2× binned. Unbinned pixel size: 9.6 Å. Scale bars: 10 μm in **B**–**C**, 5 μm in **E**–**F**, 500 nm in **G**. Figure accompanied by **Video 1**.

(***Henderson et al., 2007***), more complex eukaryotic specimens must be thinned to make them transparent to the electron beam. Two approaches have been used to examine thin samples of higher plant thylakoids. High pressure freezing followed by freeze-substitution reduces the fixation artifacts caused by conventional TEM sample preparation, enabling tomograms of plastic sections that better preserve thylakoid ultrastructure but are still limited in resolution (***Shimoni et al., 2005***; ***Austin et al., 2006***; ***Austin and Staehelin, 2011***). Alternatively, thylakoids can be isolated from crushed chloroplasts to make thin vitreous samples suitable for cryo-ET. While this approach has yielded informative high-resolution structures of protein complexes embedded in thylakoid membranes (***Daum et al., 2010***;

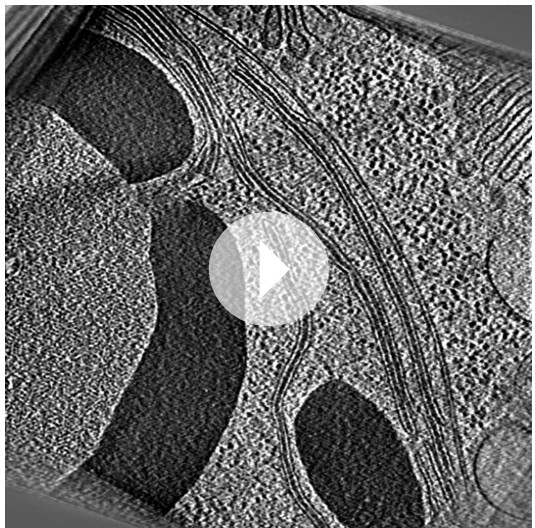

**Video 1**. Sequential sections back and forth through the tomographic volume in orthographic view. Visible structures include the pyrenoid, thylakoids, starch, and the chloroplast envelope, as well as portions of the rough endoplasmic reticulum and Golgi apparatus. The tomogram was 2× binned. Unbinned pixel size: 9.6 Å. Corresponds to *Figure 1G*.

*Kouril et al., 2011*), these membranes have been removed, both spatially and temporally, from their cellular context. Labile and transient chloroplast structures may no longer be biologically relevant in such preparations.

The ideal approach would be to thin intact vitreous specimens for cryo-ET, preserving both 3D architecture and high-resolution information. Until recently, the only method capable of thinning vitreous material was cryo-sectioning (*Hsieh et al., 2002*; *Al-Amoudi et al., 2004*), a procedure that is technically demanding, causes compression and crevasse deformations, and has a narrow range of optimal section thickness (*Al-Amoudi et al., 2005*; *Han et al., 2008*; *Bouchet-Marquis and Hoenger, 2011*). Cryo-focused ion beam (cryo-FIB) milling provides a compression-free method for thinning vitreous material to any desired thickness (*Marko et al., 2007*; *Rigort et al., 2012b*). Cryo-FIB is also more high-throughput than cryo-sectioning, as specimens can be thinned directly on the grid that will be used for cryo-ET. Cryo-FIB is thus becoming the method of choice and has already been successfully applied to a variety of eukaryotic cell types (*Hayles et al., 2010*; *Rigort et al., 2010*, *2012a*; *Strunk et al., 2012*; *Wang et al., 2012*; *Villa et al., 2013*; *de Winter et al., 2013*; *Fukuda et al., 2014*; *Hsieh et al., 2014*).

We combined cryo-FIB with cryo-ET to acquire the first in situ 3D structures of the algal chloroplast (*Figure 1*). While most structural studies have focused on terrestrial plants, aquatic organisms account for around 50% of global carbon fixation (*Field et al., 1998*; *Behrenfeld et al., 2001*), and it is estimated that nearly half of this aquatic production is performed by eukaryotic algae (*Jardillier et al., 2010*). Therefore, we chose to examine the genetically tractable unicellular green alga *Chlamydomonas reinhardtii*, the most extensively studied model organism for algal photosynthesis (*Grossman, 2000*; *Dent et al., 2001*; *Blaby et al., 2014*). Because all biological material was immobilized at the moment of whole cell vitrification, fine structures remained well preserved within their cellular context, enabling accurate measurements of thylakoid architecture, membrane invaginations, pyrenoid tubules, the eyespot, and the 3D organization of RuBisCO complexes within the pyrenoid matrix.

## Results

### Native 3D architecture of *Chlamydomonas* thylakoid membranes

*Chlamydomonas* cells are ~10 µm in diameter, and about half of their volume is occupied by a single cup-shaped chloroplast (*Figure 1A*) (*Sager and Palade, 1957*; *Gaffal et al., 1995*). Different regions of the algal chloroplast are associated with specific functions. Carbon fixation is carried out in the pyrenoid at the base of the chloroplast cup. There is evidence for chloroplast protein translation and import near the base of the cup, while the sides of the cup (known as lobes) are thought to contain mature thylakoids that perform the light reactions of photosynthesis (*Uniacke and Zerges, 2009*; *Schottkowski et al., 2012*). We acquired tomograms from a variety of cellular locations and found that all regions of the chloroplast cup share certain characteristics.

Throughout the chloroplast, thylakoids were most commonly found in stacks of variable length but uniform spacing. Thylakoid stacks had a lateral repeat distance of 22.4 ± 1.3 nm, with 4.9 ± 0.5 nm thylakoid membranes, 9.0 ± 1.4 nm thylakoid lumina, and 3.6 ± 0.5 nm interthylakoid stromal spaces (± denotes standard deviation, n = 38 thylakoids from eight stacks). Although *Chlamydomonas* thylakoid lumina were significantly wider than the thylakoid lumina reported in studies of higher plant grana (15.7–16.3 nm lateral repeat, 4.5–4.7 nm thylakoid lumina, and 3.2–3.6 nm interthylakoid stromal

spaces) (*Daum et al., 2010*; *Kirchhoff et al., 2011*), grana thylakoid lumina have been shown to expand to 9.2 nm when plants are adapted to light (*Kirchhoff et al., 2011*). In our tomograms, the number of thylakoids per stack ranged from 2 to 15, with a median of 3 thylakoids. In agreement with previous reports from other organisms (*Daum et al., 2010*; *Leforestier et al., 2012*), thylakoid lumina were less dense than the interthylakoid stromal space and the chloroplast stroma (*Figure 2*).

It is thought that all of the thylakoids within the chloroplast form one continuous compartment (*Mustárdy and Garab, 2003*; *Staehelin, 2003*). While our tomogram volumes were too confined to make a definitive assessment, 3D visualization of thylakoid architecture did indeed reveal complex interconnected networks (*Figures 3 and 4*). Thylakoid stacks frequently split along one or two of the interthylakoid stromal spaces, and the thylakoids on each side of these divisions continued and merged with other stacks (*Figure 3B*). In this manner, most thylakoid stacks were physically linked to

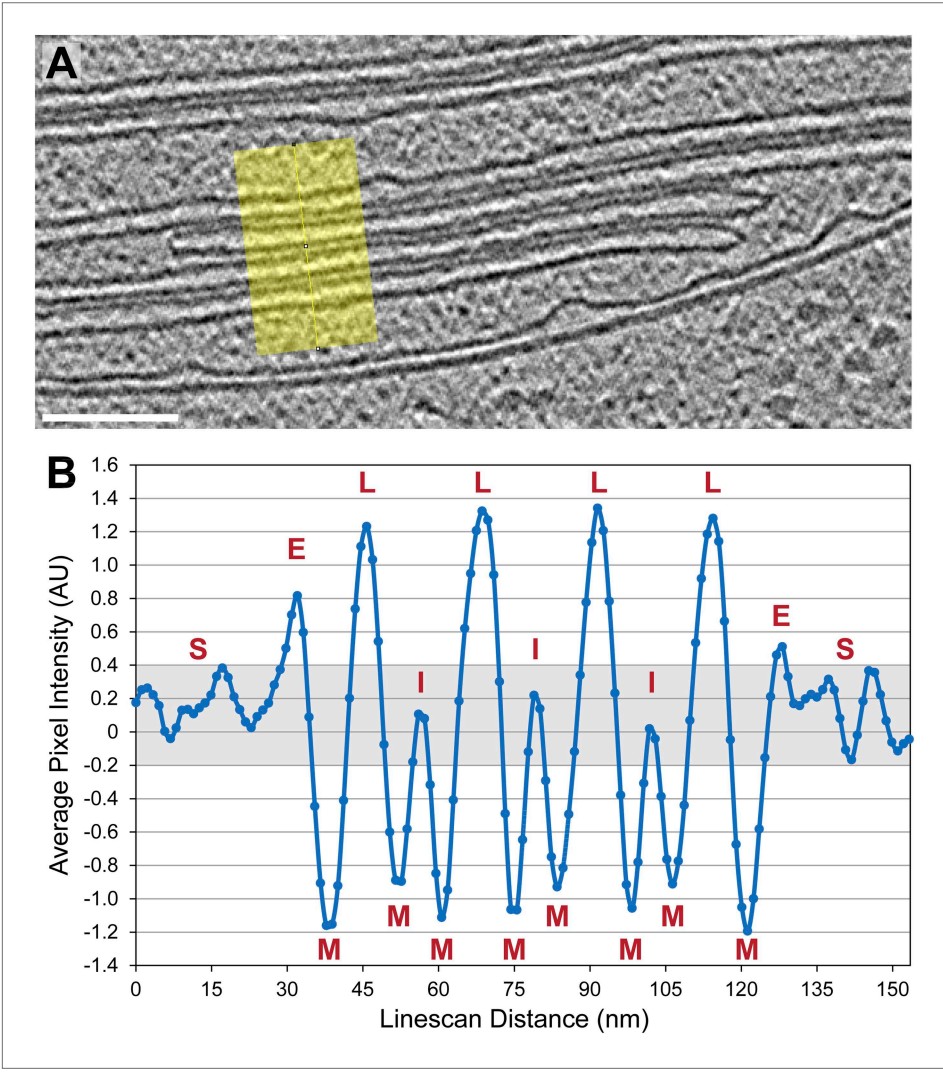

**Figure 2**. Thylakoid lumina are less dense than the interthylakoid stromal space and the chloroplast stroma. (**A**) A line scan through a 2D slice of the tomogram shown in *Figure 6A*. While tomograms were binned twice for segmentation, here the tomogram was binned only once, reducing the pixel size to 11.4 Å while decreasing contrast (unbinned pixel size: 5.7 Å). Line width: 80 pixels. Scale bar: 100 nm. (**B**) A profile from the line scan in **A** (top to bottom), plotting the average pixel intensity along the line. S: chloroplast stroma, E: edge of the thylakoid stack, M: membrane, L: thylakoid lumen, I: interthylakoid stromal space. The grey area of the plot corresponds to the range of intensity values commonly found in the chloroplast stroma. While intensities of the interthylakoid stromal spaces are within this range, the thylakoid lumina are brighter (and thus less dense). AU: arbitrary units. Line scan analysis was performed with Fiji (ImageJ, NIH).

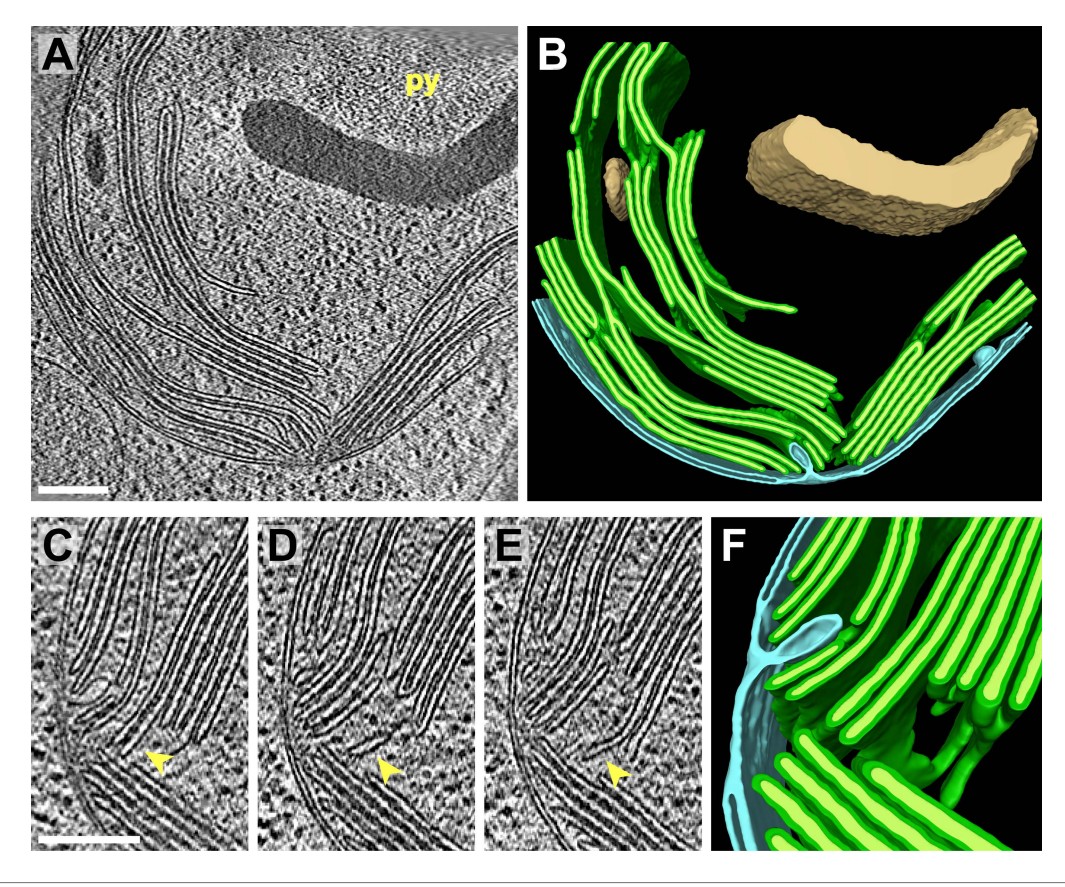

**Figure 3**. 3D cellular architecture at the base of the chloroplast cup. (**A**) A slice from a tomographic volume and (**B**) corresponding 3D segmentation showing thylakoid membranes (dark green) and lumina (light green), the chloroplast envelope (blue), and starch (tan). The dense cluster of RuBisCO complexes that make up the pyrenoid (py) can be seen in the upper right of the tomogram. (**C**–**E**) Three sequential slices from the same tomogram (rotated 90°) and (**F**) corresponding 3D segmentation, showing a thylakoid tip convergence zone. The double membrane of the chloroplast envelope has reduced definition in a small region adjacent to the converging thylakoid tips (loss of intermembrane space in **A**, **C**, and **D**). Pairs of adjacent thylakoids in a stack are often joined at intervals along their tips by 180° membrane loops. Two non-adjacent thylakoids (the second and fifth of the stack) are joined by a more topologically complex 3D loop (arrows). The tomogram was 2× binned. Unbinned pixel size: 7.1 Å. Segmented tomogram thickness: 183 nm. Scale bars: 200 nm. Figure accompanied by *Video 2*.

each other. In addition to occasional thylakoid bifurcations and mergers, thylakoid lumina were most frequently continuous with each other where the tips of two adjacent thylakoids were conjoined by a 180° loop of the thylakoid membrane (most clearly seen in *Figures 3A and 4F*). More rarely, the tips of non-adjacent thylakoids from the same stack were connected by more topologically complex looping structures (*Figure 3C–F* and light blue lumen in *Figure 10*). Like the photosynthetic membranes of cyanobacteria (*Nevo et al., 2007*; *Ting et al., 2007*), we observed fenestrations in *Chlamydomonas* thylakoids that often perforated entire stacks and may allow free diffusion of the chloroplast stroma (*Figure 5*).

## Thylakoid tip convergence zones

Thylakoids were normally oriented parallel to the chloroplast envelope, but at specific locations the thylakoid tips from multiple stacks congregated and pointed perpendicularly towards the envelope. At these thylakoid tip convergence zones (*Figure 3C–F and 4C–F*), the double membrane structure of the chloroplast envelope appeared to lose definition and in some spots was reduced to a single dense plane without an intermembrane space (*Figure 3D*). These regions also contained chloroplast envelope invaginations (*Figure 3C*) and small vesicles between the chloroplast envelope and thylakoid

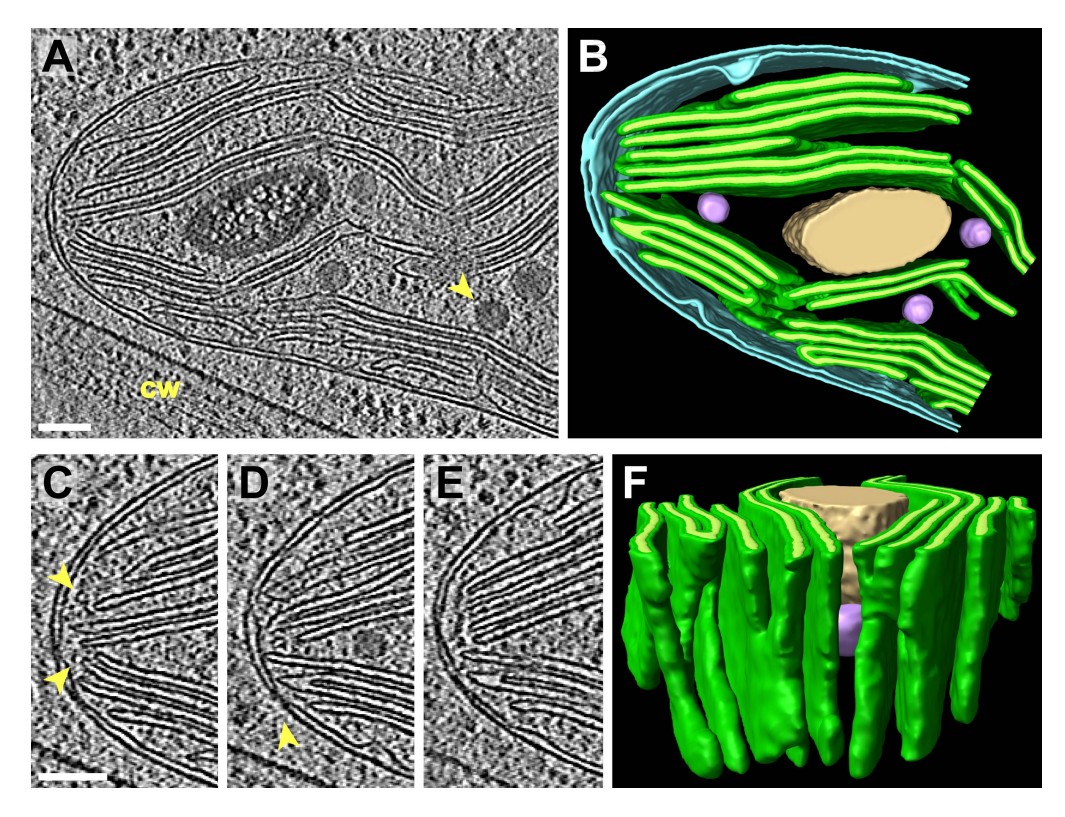

**Figure 4**. 3D cellular architecture at the rim of the chloroplast cup. (**A**) A slice from a tomographic volume and (**B**) corresponding 3D segmentation showing thylakoid membranes (dark green) and lumina (light green), the chloroplast envelope (blue), a starch granule (tan), and plastoglobules (purple, arrow in **A**). The rim of the chloroplast lobe is in close proximity to the cell wall (cw). (**C–E**) Three sequential slices from the same tomogram showing the thylakoid tip convergence zone at the chloroplast rim. Small vesicles (arrows in **C**) can be seen between the thylakoid tips and the chloroplast envelope, which loses definition in several locations (loss of intermembrane space in **C–E**). The arrow in **D** points to a circular structure in the membrane that could be a protein complex or just a single membrane bilayer. (**F**) In the corresponding 3D segmentation (which has been flipped 180˚ from **B** and tilted to face the chloroplast rim), the chloroplast envelope has been removed to reveal the interconnections between the thylakoid tips at the convergence zone. The tomogram was 2× binned. Unbinned pixel size: 5.7 Å. Segmented tomogram thickness: 126 nm. Scale bars: 100 nm. Figure accompanied by *Video 3*.

tips (*Figure 4C*). Thylakoid tip convergence zones were always present at the apical rim of the chloroplast cup (*Figure 4*), but they were also found near the base of the cup (*Figure 3*).

## Spatial relationship between inner membrane invaginations and thylakoid tips

In nearly every tomogram, we noticed invaginations that protruded from the chloroplast envelope inner membrane into the stroma (*Figures 3, 4, 6 and 7*). The density of invaginations on the membrane varied widely, with a median of 12/µm² (*Figure 8B*). Our dataset was not extensive enough to make robust statistical conclusions about the relative abundance of invaginations at the base, sides, and rim of the chloroplast cup. However, we did note that invaginations were especially prevalent near thylakoid tip convergence zones and at the base-lobe junction (*Figure 7*), where the base and the sides of the chloroplast cup meet.

While invaginations were found everywhere along the chloroplast envelope inner membrane, we noticed that many invaginations occurred in close proximity to thylakoid tips (*Figure 6A–D and 7*). In our dataset, 34% of the invaginations were positioned less than 40 nm from a thylakoid tip, and 64% were found within 100 nm of a tip (*Figure 8D*). Thus, inner membrane invaginations may provide a route for the exchange of lipids and proteins between the chloroplast envelope and thylakoid

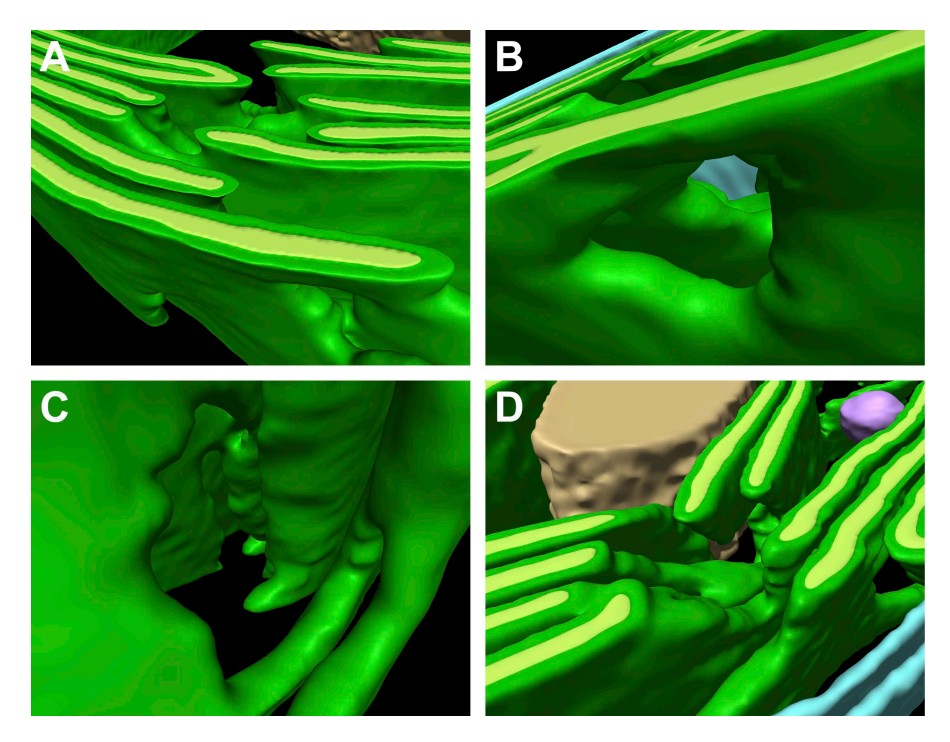

**Figure 5**. Thylakoid fenestrations. Alternate views of the 3D segmentations from (**A**–**C**) *Figure 3* and (**D**) *Figure 4* showing fenestrations in thylakoid stacks. In each example, there are holes at similar positions in multiple adjacent thylakoids, enabling free diffusion of stromal components from one side of the thylakoid stack to the other. In **A** and **D**, the fenestrations are not completely within the limited thickness of the FIB lamella, and hence appear like indentations in the top surfaces of the segmented thylakoid volumes.

compartments. Indeed, we observed a few direct connections between thylakoid tips and the chloroplast envelope inner membrane (*Figure 6E–H*). These connections were rare, which could help maintain the distinct identities of each membrane and preserve the high proton concentration within the thylakoids.

## Starch structures and plastoglobules

The cluster of RuBisCO complexes in the pyrenoid was encased in a starch sheath (*Figures 1G, 3A–B and 9A–B*), and non-spherical starch granules were found throughout the chloroplast (*Figures 1G, 3A–B, 4A–B and 7A–B*). In addition to starch, we observed smaller 64 ± 15 nm spherical plastoglobules (n = 20 globules from 9 tomograms) (*Greenwood et al., 1963*; *Ohad et al., 1967b*) that were always closely associated with thylakoid membranes (*Figure 4*). Higher plant plastoglobules were previously shown by freeze-fracture to share a half-lipid bilayer with thylakoid membranes (*Austin et al., 2006*). Given the caveat that the resolution of our CCD-acquired tomograms did not reveal the two layers of lipid bilayers, *Chlamydomonas* plastoglobules appeared to be discontinuous from thylakoid membranes, although punctate contacts were sometimes observed (*Figure 4A,D*). Additionally, while higher plant plastoglobules are frequently localized to areas of high membrane curvature, where they are proposed to form (*Austin et al., 2006*), *Chlamydomonas* plastoglobules were primarily found adjacent to flat thylakoid sheets (*Figures 4, 7A–B and 9A*). Although the composition of *Chlamydomonas* plastoglobules is currently unknown (*Goodson et al., 2011*), they are believed to contain lipids and thylakoid components similar to the plastoglobules of higher plants (*Steinmüller and Tevini, 1985*; *Ytterberg et al., 2006*). Consistent with this hypothesis, *Chlamydomonas* plastoglobule structure remained intact at electron doses that did not affect lipid membranes but damaged the radiation-sensitive starch granules (seen by bubbling within the granules, *Figure 4A*). Such a 'bubblegram' approach can discriminate between chemically distinct entities in cryo-tomograms (*Wu et al., 2012*).

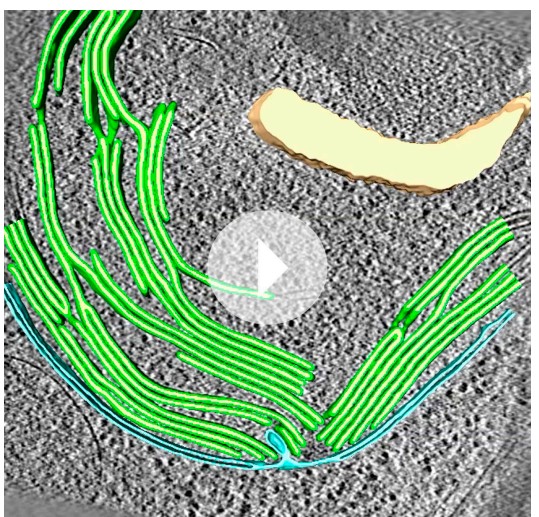

**Video 2**. Sequential sections back and forth through the tomographic volume in orthographic view, followed by reveal and tour of the 3D segmentation in perspective view. The tour focuses on the thylakoid tip convergence zone and the topologically complex looping structure connecting two non-adjacent thylakoid tips. The tomogram was 2× binned. Unbinned pixel size: 7.1 Å. Corresponds to **Figure 3**.

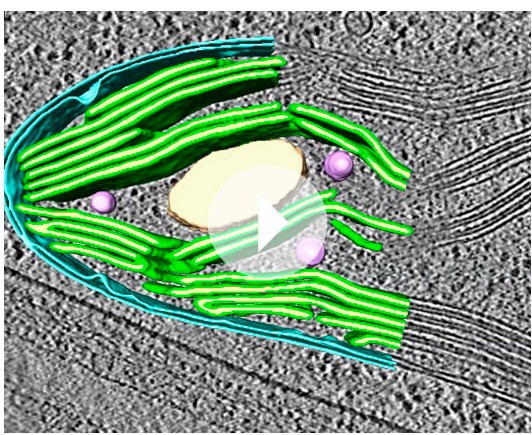

**Video 3**. Sequential sections back and forth through the tomographic volume in orthographic view, followed by reveal and tour of the 3D segmentation in perspective view. The tour focuses on the thylakoid tip convergence zone at the rim of the chloroplast cup. Halfway through the tour, the chloroplast envelope is removed to reveal the interconnections between adjacent thylakoid tips. The tomogram was 2× binned. Unbinned pixel size: 5.7 Å. Corresponds to **Figure 4**.

## Pyrenoid tubules form conduits between thylakoid stacks and the pyrenoid

Membrane tubules extend from the thylakoids into the pyrenoid through fenestrations in the starch sheath (*Sager and Palade, 1957*; *Ohad et al., 1967b*). We aimed to better understand the function of these intriguing structures by visualizing their native 3D architecture (*Figures 9–11*). Thylakoid stacks often changed direction (*Figure 9A–D*) as their sheets merged with each other and narrowed into cylindrical pyrenoid tubules (*Figure 10*), which had a mean diameter of 107 ± 26 nm (n = 46 tubules from 10 tomograms). A 5–10 nm RuBisCO-free space was frequently observed surrounding the tubules (*Figure 9E,G*). Deeper within the pyrenoid, pyrenoid tubules began to twist (*Figure 9A*, top tubule), sometimes abruptly changed direction by >90° (*Figure 9B*, bottom tubule), and eventually lost their cylindrical structure as the tubules converged at the pyrenoid center, forming elaborate interconnected networks of variably shaped smaller membranes (*Figure 9B*).

Pyrenoid tubules have previously been reported to contain 'ridges' or 'infoldings' along the inner surface of the tubule membrane (*Ohad et al., 1967a*). Interestingly, in high magnification cryotomograms these structures were revealed to be smaller tubules (*Figure 9E–H* and *Figure 9—figure supplement 1*), which we have termed pyrenoid minitubules. Each pyrenoid tubule encompassed between two and eight minitubules, with a median of 5. In cross-section, the minitubules had an oval or bean-like shape (*Figure 9E,G*) with a long-axis diameter of 21.6 ± 1.7 nm and a short-axis diameter of 13.4 ± 1.1 nm, including the membrane (n = 22 minitubules within four pyrenoid tubules). The minitubules enclosed hollow lumina that were 3.5 ± 0.5 nm wide along the short-axis, comparable in width to the interthylakoid stromal space.

Segmentation of the stack-to-tubule transition showed that the lumina of pyrenoid tubules were continuous with the lumina of thylakoids, while the lumina of pyrenoid minitubules were continuous with the stroma in the interthylakoid space (*Figure 10*). Minitubule membranes were derived from the coalescence of two opposed membranes of adjacent thylakoids in a stack (*Figure 10C–F*). Consequently, minitubule membrane topology is 'inside-out' relative to the membranes of pyrenoid tubules. At the periphery of the pyrenoid, minitubules were completely enveloped within the surrounding pyrenoid tubule membrane (*Figure 9E–H*). As pyrenoid tubules proceeded towards the center of the pyrenoid, minitubules emerged from the pyrenoid tubules, exposing their stromal lumina to the pyrenoid matrix (*Figure 11A–E*). Thus, pyrenoid minitubules form continuous conduits between

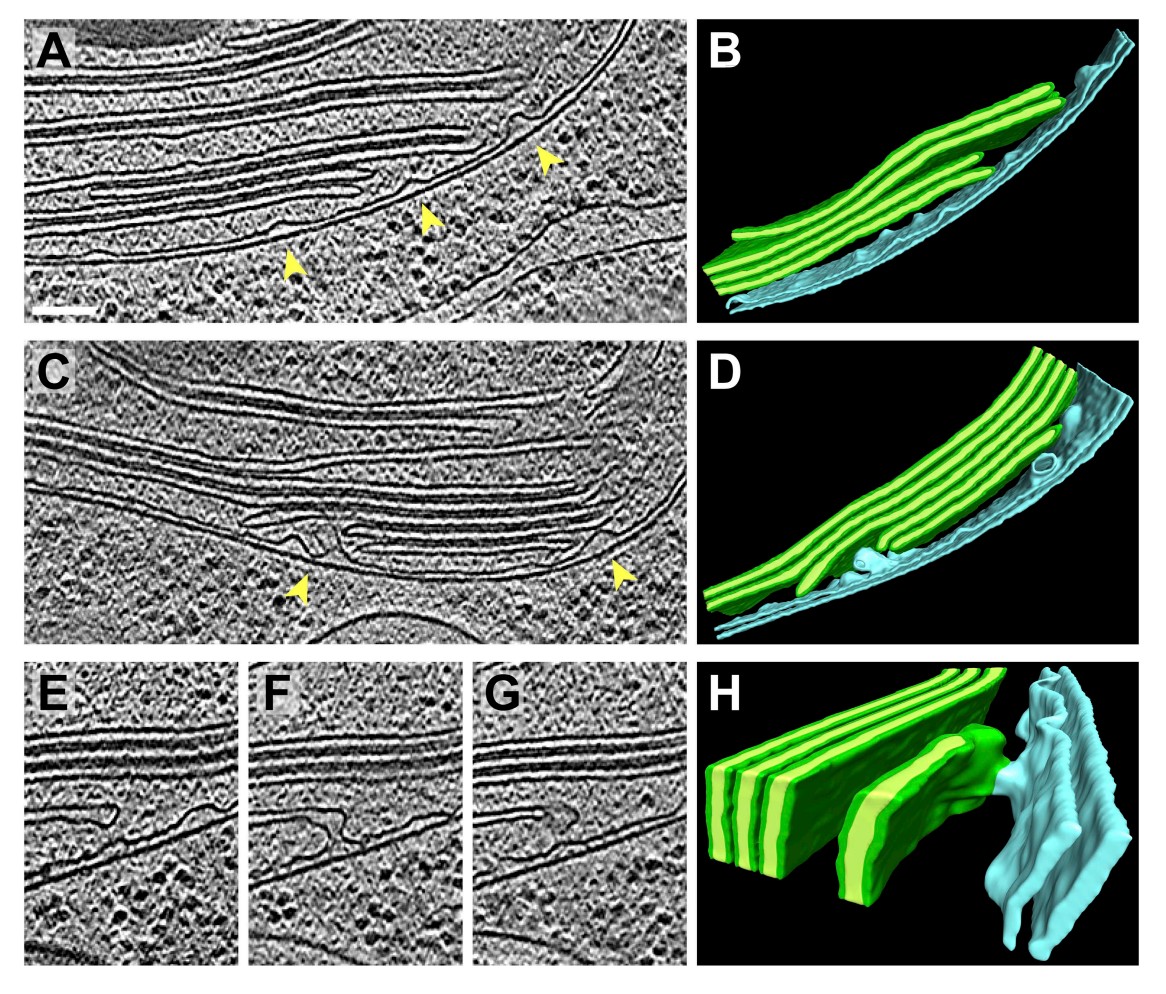

**Figure 6**. Spatial relationship between chloroplast envelope inner membrane invaginations and thylakoid tips. (**A** and **C**) Slices from tomographic volumes and (**B** and **D**) corresponding 3D segmentations showing thylakoid membranes (dark green), thylakoid lumina (light green), and the chloroplast envelope (blue). Inner membrane invaginations (arrows) are frequently found in close proximity to the tips of thylakoid sheets. (**E–G**) Three sequential slices through a tomogram and (**H**) corresponding 3D segmentation, where a region of thylakoid tip (~25 nm high and ~20 nm wide) is connected to the chloroplast envelope inner membrane. The tomograms were 2× binned. Unbinned pixel size: 5.7 Å. Segmented tomogram thickness: 137 nm in **B**, 135 nm in **D**, 69 nm in **H**. Scale bar: 100 nm.

the interthylakoid stromal space and the RuBisCO complexes in the pyrenoid matrix. Further inside the pyrenoid, pyrenoid tubules merged with each other, producing additional internal membrane structures with luminal spaces that were also continuous with the pyrenoid matrix (**Figure 11F–J**).

## 3D organization of RuBisCO complexes within the pyrenoid

RuBisCO accounts for over 90% of the pyrenoid's protein content (**Holdsworth, 1971**). Such homogeneity suggests that the pyrenoid may exhibit higher-order organization, and indeed, 'crystalline' packing of RuBisCO complexes has been described in a few algal pyrenoids (**Holdsworth, 1968**; **Kowallik, 1969**) as well as bacterial carboxysomes (**Schmid et al., 2006**; **Iancu et al., 2007**). However, the pyrenoids of most species, including those found in *Chlamydomonas*, appear amorphous (**Griffiths, 1970**; **Meyer et al., 2012**). Since these conclusions about pyrenoid architecture are based on 2D EM of plastic sections, we decided to investigate whether RuBisCO complexes were ordered within our native-state tomographic volumes of the *Chlamydomonas* pyrenoid.

Upon visual inspection, the dense matrix of RuBisCO complexes did not obviously appear to be arranged in a crystalline lattice (**Figure 12A**). However, symmetry-free subtomogram averaging of

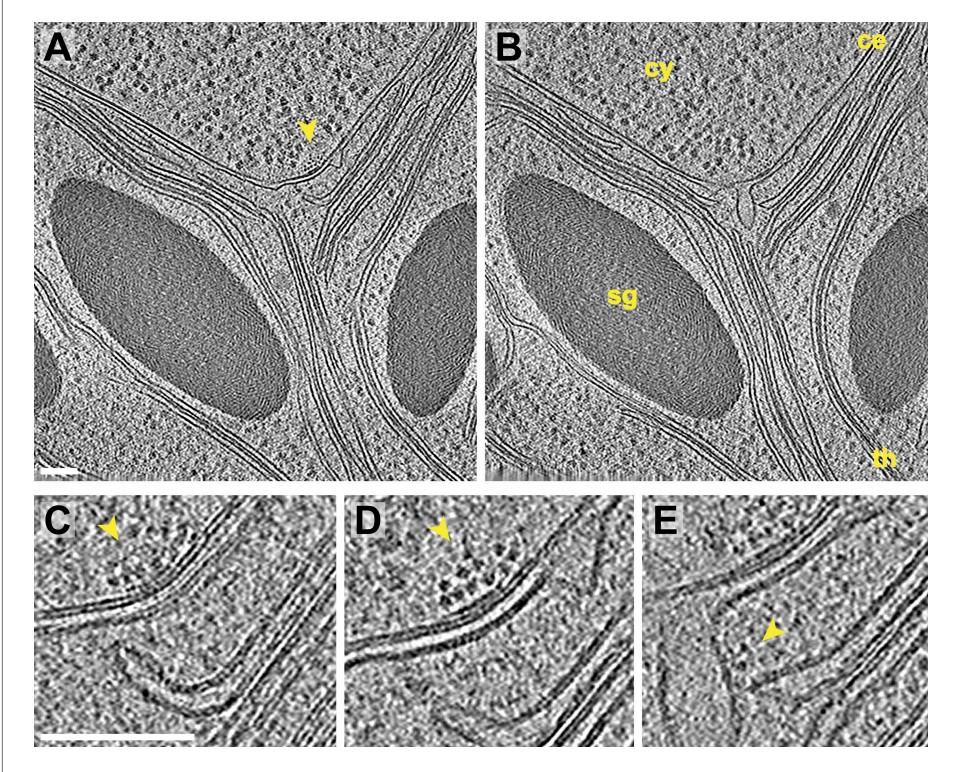

**Figure 7**. Inner membrane invaginations and membrane-associated complexes at the chloroplast base-lobe junction. (**A–B**) Two sequential slices through a tomogram acquired with a direct electron detector, showing a junction between the base and the side (lobe) of the chloroplast cup. At the bend in the chloroplast envelope, an array of small complexes (arrow) is associated with the cytoplasmic surface of the envelope's outer membrane, between two invaginations in the inner membrane. The cytoplasm (cy), chloroplast envelope (ce), starch granules (sg), and thylakoids (th) are labeled. (**C–E**) Three sequential slices through the same tomographic volume, showing a magnified view of the chloroplast envelope at the base-lobe junction. **D** and **E** correspond to the views in **A** and **B**, respectively. Separate halves of lipid bilayers were fairly well resolved along relatively flat membranes. A thylakoid tip (**C**) bends ~90° to point towards the chloroplast envelope and (**E**) nearly contacts a large inner membrane invagination. Small complexes (arrows) are clustered in an array on the outer membrane and are also associated with the inner membrane invagination close to the thylakoid tip. These particles each have a diameter of ~6 nm. For comparison, the diameter of the $F_1$ ATP synthase subunit is ~9 nm and the diameter of the fully assembled RuBisCO complex is ~12 nm (note that the RuBisCO large subunit and six ATP synthase components are encoded by the chloroplast genome and thus, are not imported). The tomogram was 3× binned in **A–B** and 2× binned in **C–E**. Unbinned pixel size: 3.4 Å. Scale bars: 100 nm.

45.6 nm$^3$ subvolumes revealed that the RuBisCO complexes were actually hexagonally packed (***Figure 12B***). The intensity of the average was somewhat asymmetric due to local deviations from a perfectly packed crystal. Nevertheless, each of the 13 densities, corresponding to a unit cell of a RuBisCO complex and its neighbors, were located close to the ideal centers of hexagonal close packing (***Figure 12B***, blue diamonds). Furthermore, the average clearly showed the A-B-A layered arrangement of complexes that is a hallmark of hexagonal close packing (***Figure 12B***, bottom right).

We determined the relative center positions of the RuBisCO complexes via an independent high-precision method (***Figure 13***). After localizing each RuBisCO complex in our tomograms by template matching, the nearest neighbors of each RuBisCO were overlaid in a point cloud, which again showed a clear A-B-A hexagonal arrangement (***Figure 13B***). The center positions of each neighbor in the point cloud were calculated by k-means clustering (***Figures 12C and 13***, red diamonds). We generated ideal hard sphere models and found that hexagonal close packing had a smaller root-mean-squared deviation (RMSD) from our point cloud center positions than cubic close packing or body-centered cubic packing (***Figure 12D***). Additionally, hexagonal close packing with a spherical diameter of 15.05 nm yielded a

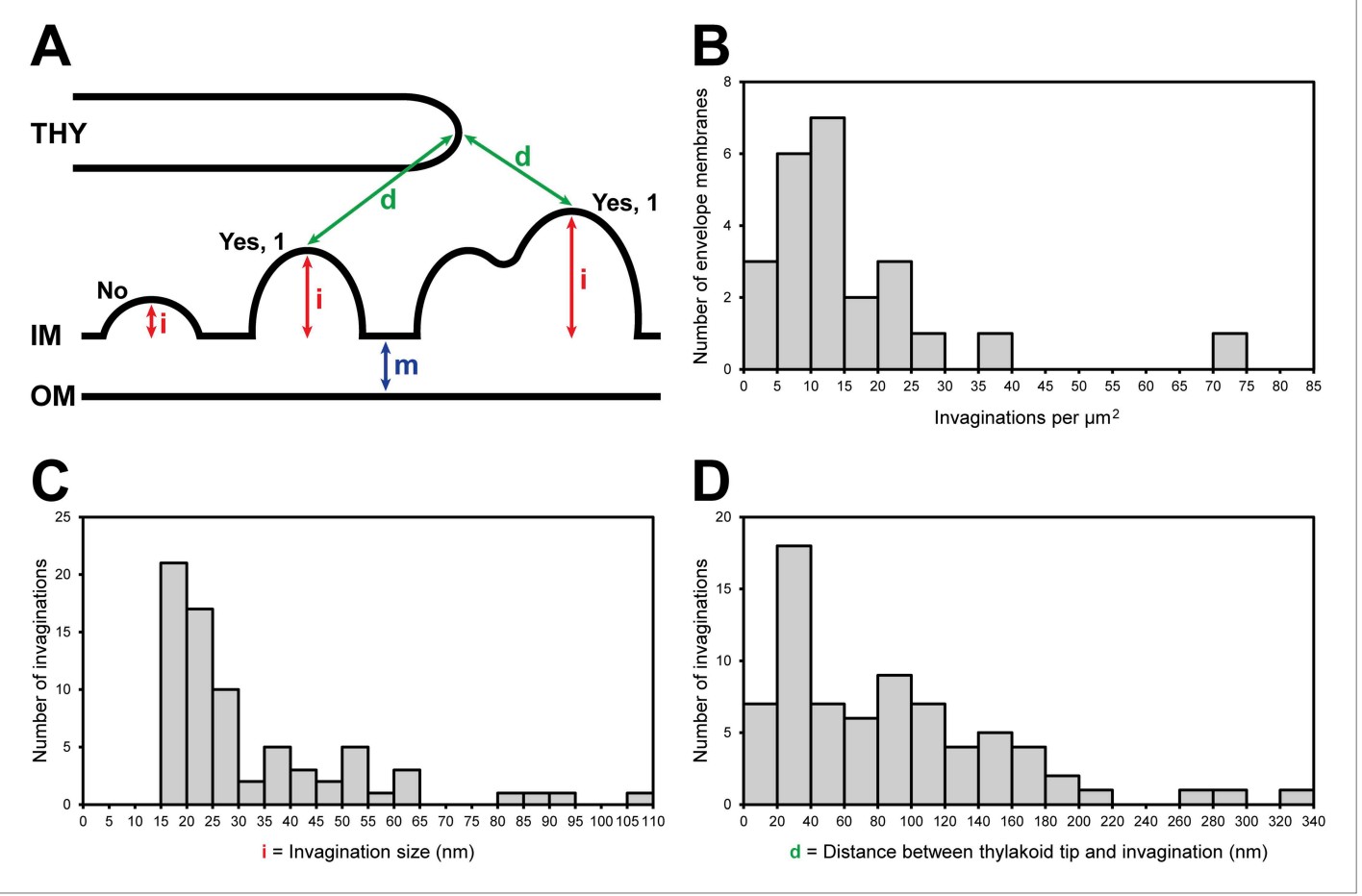

**Figure 8**. Chloroplast envelope inner membrane invagination size, density on the inner membrane, and distance from thylakoid tips. (**A**) Diagram illustrating how chloroplast envelope inner membrane (**B**) invagination density, (**C**) invagination size, and (**D**) distance between invagination and thylakoid tip were quantified. Invaginations were counted if the distance they projected into the stroma (i, invagination size) was greater than the width of the chloroplast envelope's double membrane (m). For this reason, there are no invaginations less than 15 nm in size plotted in **C**. Invaginations with multiple tips were counted as single invaginations. The distance between thylakoids and invaginations (**D**) was measured as the shorted straight line through the 3D volume between the tip of an invagination and the tip of the nearest thylakoid.

unique minimum RMSD, indicating that RuBisCO complexes within the pyrenoid have ~15 nm between their centers (***Figure 12D***). Given the dimensions of RuBisCO, complexes positioned 15 nm apart would have a 2–4.5 nm space between their surfaces (***Figure 12E***).

## 3D architecture of the eyespot

The eyespot apparatus is a specialized region of the chloroplast and adjacent plasma membrane that controls phototaxis by detecting the intensity and direction of light (***Kreimer, 2009***). Viewed by cryo-ET, the eyespot globules were similar in appearance to plastoglobules dispersed throughout the chloroplast (***Figure 14A***), although they were slightly larger (***Figure 14G***) and densely packed along two adjacent thylakoid membranes. The two arrays of eyespot globules, the associated thylakoids, and the bordering chloroplast envelope were all slightly concave relative to the plasma membrane (***Figure 14B***). However, *Chlamydomonas* eyespots with straight or convex globule arrays have frequently been observed in plastic section TEM images (***Kreimer, 2009***), indicating that eyespot curvature may be variable. The eyespot globules are proposed to reflect incident light coming from outside the cell back to channelrhodopsin photoreceptors in the overlying plasma membrane, while simultaneously absorbing light passing through the cell from other directions (***Foster and Smyth, 1980***; ***Kreimer and Melkonian, 1990***). Concave eyespot architecture may enhance reflection, as it has been shown to focus reflected light in a different species of green algae (***Kreimer and Melkonian, 1990***).

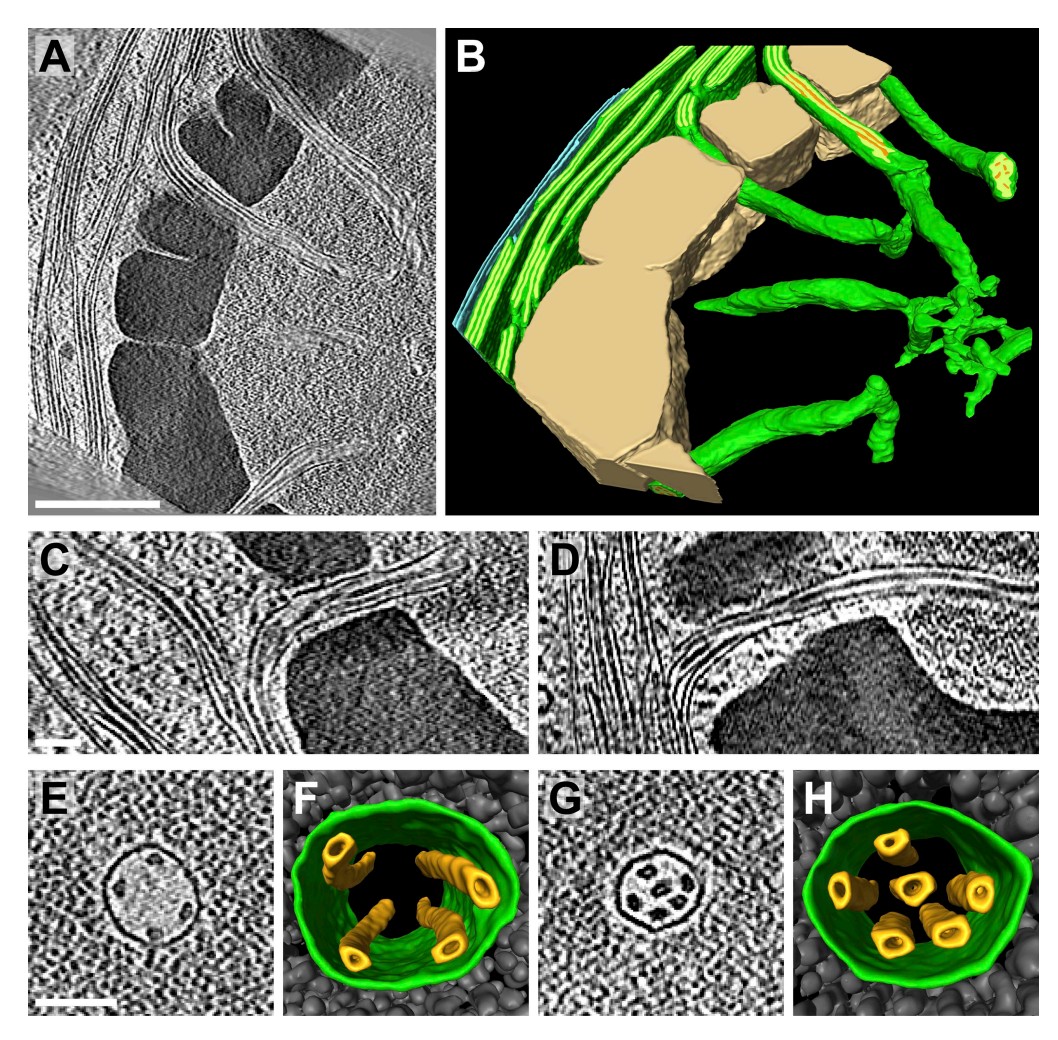

**Figure 9**. Pyrenoid tubules connect thylakoids with the pyrenoid. (**A**) A slice from a tomographic volume and (**B**) corresponding 3D segmentation showing thylakoid membranes (dark green), thylakoid lumina (light green), minitubules (orange), the chloroplast envelope (blue), and the pyrenoid starch sheath (tan). Cylindrical pyrenoid tubules extend from thylakoid stacks into the center of the pyrenoid through fenestrations in the starch sheath. Once inside the pyrenoid, the tubules twist (top tubule in **A**), can abruptly change direction by ~90° (bottom tubule in **B**), lose their cylindrical geometry, and narrow to form an interconnected membrane network at the pyrenoid center. (**C–D**) Slices from two additional tomograms detailing how pyrenoid tubules extend from thylakoid stacks. Thylakoid lumina are continuous with pyrenoid tubule lumina. (**E** and **G**) Slices from higher magnification tomograms of the pyrenoid matrix and (**F** and **H**) corresponding 3D segmentations showing cross-sections of pyrenoid tubules (green) surrounded by RuBisCO complexes (grey). Multiple smaller minitubules (orange) are bundled within each pyrenoid tubule. The tomograms were 2× binned. Unbinned pixel size: 9.6 Å in **A–D**, 5.7 Å in **E–H**. Segmented tomogram thickness: 229 nm in **B**, 165 nm in **F**, 108 nm in **H**. Scale bars: 500 nm in **A**, 100 nm in **C**, **D**, **E**, and **G**. Figure accompanied by *Videos 4 and 5*.

The following figure supplement is available for figure 9:

**Figure supplement 1**. Longitudinal views of pyrenoid minitubules.

---

We observed a cluster of unilamellar and bilamellar vesicles embedded between the cell wall and the rhodopsin-containing region of the plasma membrane (*Figure 14D–F*). To our knowledge, such structures have not been previously described, although voids in the cell wall near the eyespot have been seen in conventional plastic sections (G Kreimer and S Geimer, personal communication, May 2014). Comparing all of our tomograms containing the cell wall, these vesicles were only observed

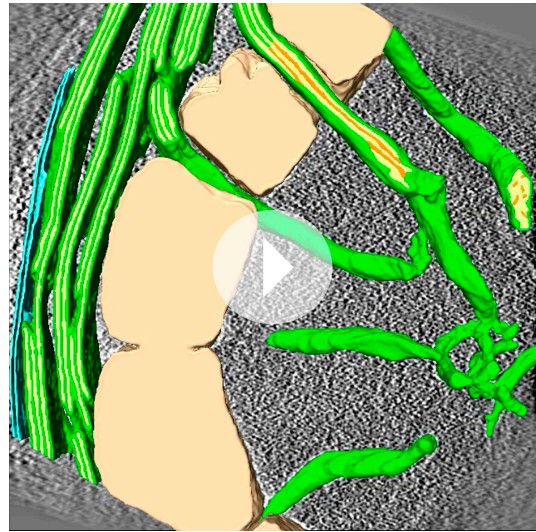

**Video 4.** Sequential sections back and forth through the tomographic volume in orthographic view, followed by reveal and tour of the 3D segmentation in perspective view. The tour focuses on the pyrenoid tubules, which fenestrate the starch sheath and transition from cylindrical channels to an interconnected network of smaller membranes at the center of the pyrenoid. The tomogram was 2× binned. Unbinned pixel size: 9.6 Å. Corresponds to **Figure 9A–B**.

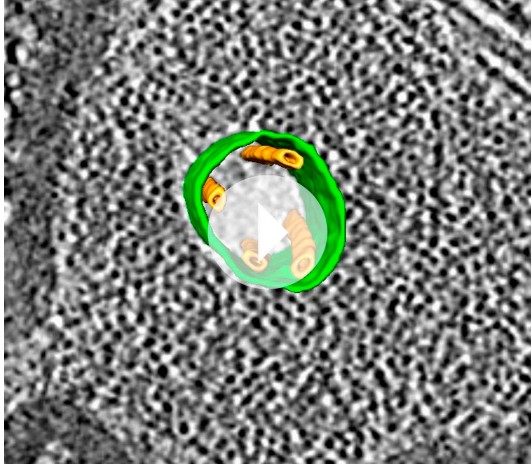

**Video 5.** Sequential sections back and forth through the tomographic volume in orthographic view, followed by reveal and tour of the 3D segmentation in perspective view. The segmentation shows the fine structure of a pyrenoid tubule that encases four minitubules. The tomogram was 2× binned. Unbinned pixel size: 5.7 Å. Corresponds to **Figure 9E–F**.

adjacent to the eyespot apparatus, implying that they may be related to eyespot function.

## Discussion

### The native 3D architecture of *Chlamydomonas* thylakoids raises molecular questions

Unlike higher plants, algal chloroplasts do not have a distinct division between grana and stroma thylakoids. Comparing the structures of algal and higher plant thylakoid stacks may shed light on the evolution of photosynthetic systems. The *Chlamydomonas* interthylakoid stromal space (3.6 nm) is comparable to the grana interthylakoid stromal space (3.2–3.6 nm) (*Daum et al., 2010*; *Kirchhoff et al., 2011*), indicating that the molecular mechanisms that hold two algal or higher plant thylakoid membranes together may be similar. This attraction is proposed to involve van der Waals and electrostatic interactions between membrane surfaces (*Barber and Chow, 1979*; *Chow et al., 2005*), perhaps mediated by light-harvesting complexes bound to photosystem II (PSII) (*Daum et al., 2010*). The oxygen-evolving complex (OEC) of PSII projects 4.5 nm into the thylakoid lumen (*Ferreira et al., 2004*), a comparable distance to the 4.5–4.7 nm width of grana thylakoid lumina measured for isolated chloroplasts and leaves of dark-adapted plants (*Daum et al., 2010*; *Kirchhoff et al., 2011*). Thus, OECs on opposite sides of these narrow lumina must interdigitate due to spatial constraints (*Daum et al., 2010*), and indeed, it seems likely that the size of the OEC determines the minimum grana lumen width. Our *Chlamydomonas* cells, which were grown in constant light until the moment of vitrification, had 9 nm thylakoid lumina. Interestingly, this may be a result of light conditions rather than an algal characteristic, as leaves that were harvested from light-exposed plants and immediately cryo-immobilized had 9.2 nm grana lumina (*Kirchhoff et al., 2011*). Thus, PSII complexes in *Chlamydomonas* thylakoids and light-exposed higher plant grana should experience fewer limitations to their mobility, and their OECs may directly face each other instead of being interdigitated. Examination of dark-adapted *Chlamydomonas* cells would reveal whether the light-induced change in thylakoid lumen width is conserved between algae and higher plants.

While much research in higher plants has focused on the mechanisms that organize thylakoid stacks, little is known about how proteins orchestrate the dramatic membrane curvature seen in our tomograms, including the 180˚ loops between thylakoid tips, the transition from flat thylakoid

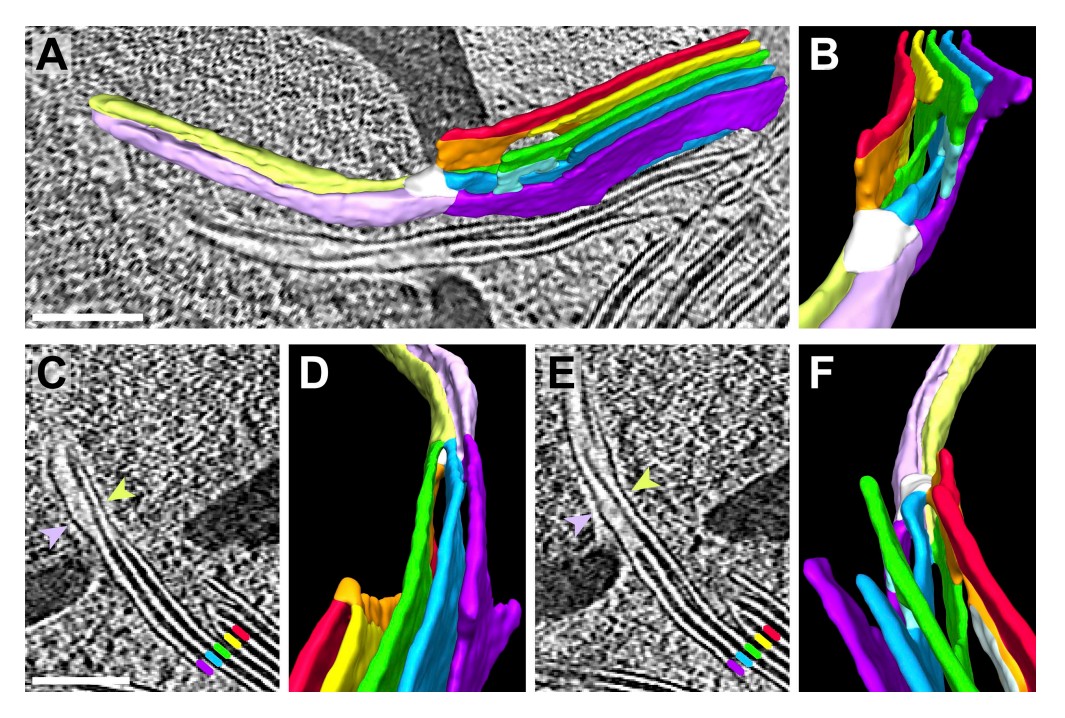

**Figure 10**. Pyrenoid minitubule lumina are continuous with the interthylakoid stromal space. Slices from a tomographic volume and different views of the corresponding 3D segmentation showing the transition from a thylakoid stack to a pyrenoid tubule. To ease visualization, the continuous luminal space has been segmented rather than the membranes. Colors indicate lumen connectivity. Lumina of the thylakoid stack are colored red, yellow, green, blue, and purple. The red and yellow lumina merge to form the orange lumen, which then connects to the green lumen. The light blue lumen bridges the green lumen with both the blue and purple lumina. All lumina converge in the white 'neck' region, where the thylakoids transition into a pyrenoid tubule. While the pyrenoid tubule lumen is continuous, the light green half is more directly connected to the orange and green thylakoid lumina, and the lavender half is more directly connected to the blue and purple lumina. The two indentations between the light green and lavender halves correspond to minitubules within the pyrenoid tubule. (**A**) Side view of the segmentation superimposed above a tomographic slice and (**B**) end-on view of the segmentation, rotated ~90˚ from **A**. (**C**–**D**) The interior of one pyrenoid minitubule is continuous with the interthylakoid stromal space between the purple and blue lumina. (**E**–**F**) The interior of the second pyrenoid minitubule is continuous with the interthylakoid stromal space between the blue and green lumina. The yellow lumen has been removed in **F** to provide an unobstructed view. The view in **D** is inverted 180˚ from **F** to show both the top and bottom faces of the segmentation. In **C** and **E**, lines and arrows indicate the corresponding colored thylakoid lumina and halves of the pyrenoid tubule, respectively. The tomogram was 2× binned. Unbinned pixel size: 9.6 Å. Segmented tomogram thickness: 187 nm. Scale bars: 200 nm. Figure accompanied by *Video 6*.

stacks to cylindrical pyrenoid tubules, and the very thin pyrenoid minitubules. One good candidate for the 180˚ loops is CURT1, an evolutionarily conserved family of proteins that oligomerize, tubulate membranes, and are enriched at the grana margins of higher plants (*Armbruster et al., 2013*; *Pribil et al., 2014*). Recent technical advancements in correlative microscopy and cryo-ET will enable the in situ localization and structural characterization of the protein complexes at curved thylakoid and pyrenoid tubule membranes.

## Chloroplast envelope inner membrane invaginations and thylakoid tip convergence zones have implications for thylakoid biogenesis

While the thylakoid and chloroplast envelope membranes are widely believed to be independent systems, thylakoids have a similar lipid composition to the chloroplast envelope inner membrane (*Kelly and Dormann, 2004*) and require the import of proteins encoded by the nuclear genome (*Green, 2011*). Thus, transport between these systems must be possible. Over one third of the inner

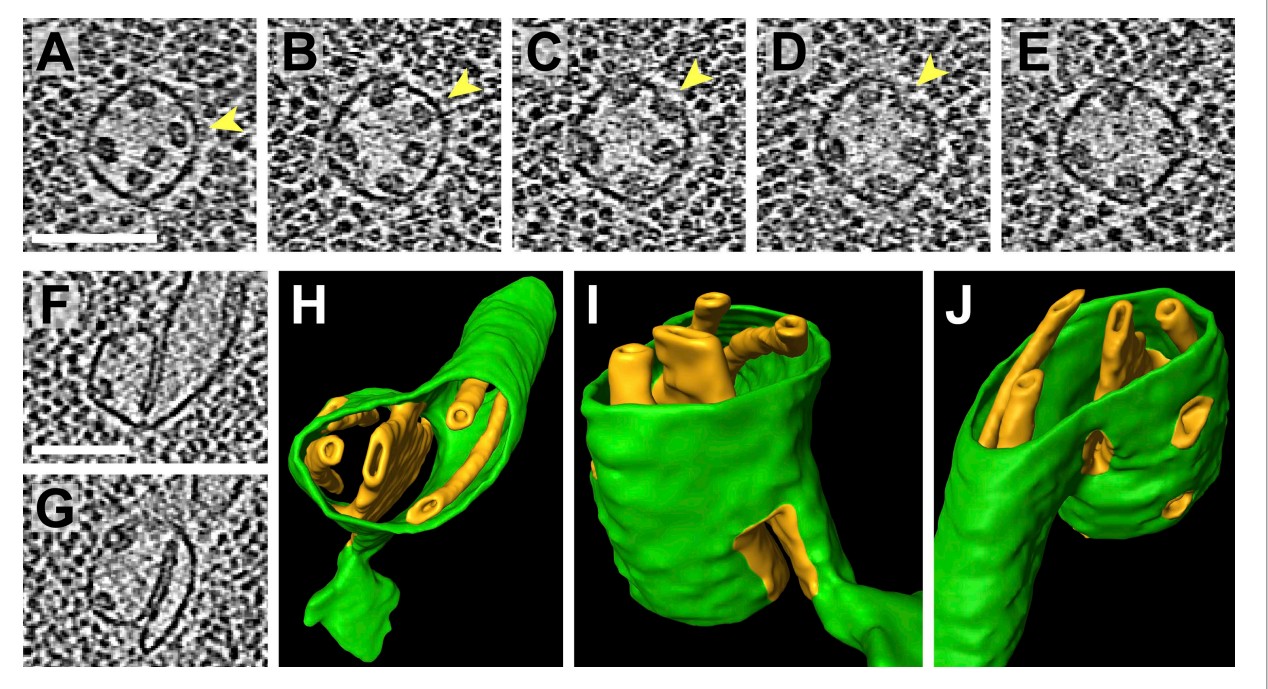

**Figure 11**. Pyrenoid minitubule lumina are continuous with the pyrenoid matrix. (**A–E**) Sequential slices through a tomographic volume, following a pyrenoid tubule as it proceeds towards the center of a pyrenoid. In **C** and **D**, the minitubule marked with an arrow terminates by merging with the surrounding pyrenoid tubule, exposing the minitubule lumen to the pyrenoid matrix, which is packed with RuBisCO complexes. The tomogram in **A–E** was acquired with a direct electron detector. (**F–G**) Two slices from a tomogram and (**H–J**) corresponding 3D segmentation showing the merger of two pyrenoid tubules (green) near the center of a pyrenoid. In addition to the minitubules (orange) within each of the pyrenoid tubules, there is a larger internal membrane structure (orange) at the zone where the two pyrenoid tubules meet. This membrane structure is more sheet-like than the minitubules but also encloses a lumen that is continuous with the pyrenoid matrix. **F** and **G** show that the lumen of this membrane structure opens to the pyrenoid matrix on both sides of the merging pyrenoid tubules. In **H** and **I**, a non-cylindrical membrane (green) can be seen protruding from the merging tubules. The tomograms were 2× binned. Unbinned pixel size: 4.2 Å in **A–E**, 5.7 Å in **F–J**. Segmented tomogram thickness: 130 nm. Scale bars: 100 nm.

membrane invaginations in our tomograms were located within 40 nm of a thylakoid tip (*Figure 8D*), less than the width of two stacked thylakoids. Taken together with the rare direct connections we observed between the inner membrane and thylakoid tips (*Figure 6E–H*), it is likely that there is a transport mechanism for the exchange of lipids and proteins between these two compartments. Consistent with this idea, there is growing evidence that vesicular transport from the chloroplast envelope inner membrane is required for thylakoid biogenesis (*Kroll et al., 2001*; *Garcia et al., 2010*; *Nordhues et al., 2012*). If such a transport pathway exists, it will be interesting to understand how the specific targeting to thylakoid tips is achieved (*Khan et al., 2013*).

The thylakoid tip convergence zones that we described in this study (*Figures 3 and 4*) are likely sites where there is prolific exchange of lipids and proteins between the thylakoid and chloroplast envelope membranes. Although thylakoid tip convergence zones at the base and rim of the chloroplast cup appear morphologically similar, the question remains as to whether they all perform the same function. Recent work suggests that *Chlamydomonas* thylakoid biogenesis may be spatially regulated (*Uniacke and Zerges, 2009*; *Schottkowski et al., 2012*). In the model proposed by these studies, thylakoid proteins encoded by the chloroplast genome are locally translated at the base of the chloroplast cup in a region surrounding the pyrenoid called the T-zone (*Uniacke and Zerges, 2007*). At the base-lobe junction, where the base and sides of the cup meet, nuclear-encoded proteins are imported and assembled with the chloroplast-encoded proteins into macromolecular complexes in the thylakoid membranes. Finally, mature thylakoids in the sides of the cup carry out the photosynthetic light reactions. If this model is correct, the thylakoid tip convergence zones at the base and rim of the chloroplast cup likely perform different functions. The base of the cup is located close to the ER and Golgi, as well as many cytoplasmic ribosomes (*Figure 1G*), and thus convergence zones at the

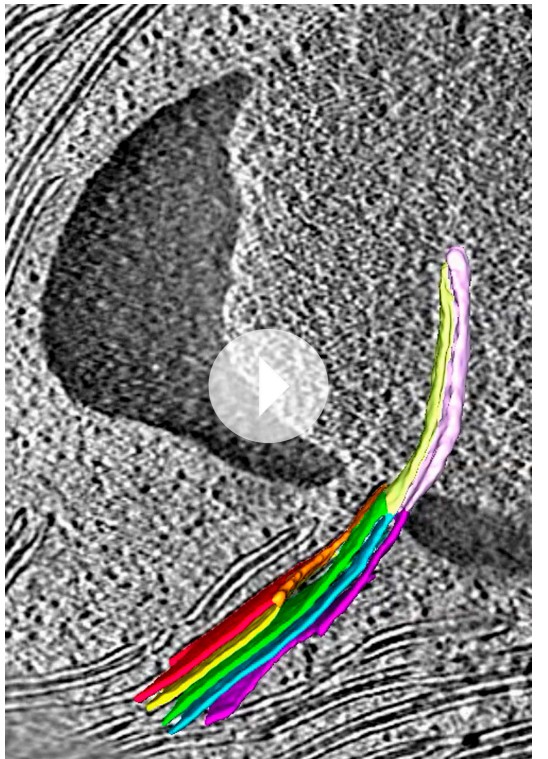

**Video 6**. Sequential sections back and forth through the tomographic volume in orthographic view, followed by reveal and tour of the 3D segmentation in perspective view. The tour first shows how one pyrenoid minitubule (indentation between the light green and lavender halves of the pyrenoid tubule) originates from the space between the blue and purple thylakoids (**Figure 10C–D**). The segmentation is then flipped over to reveal how a second minitubule originates from the space between the green and blue thylakoids (**Figure 10E–F**). The tomogram was 2× binned. Unbinned pixel size: 9.6 Å. Corresponds to **Figure 10**.

base-lobe junction are ideally situated for the import of new proteins. Indeed, in this junction region we observed small complexes associated with both the chloroplast envelope outer membrane and inner membrane invaginations, indicating that these complexes may be undergoing import (**Figure 7**).

Unlike thylakoid tip convergence zones near the base of the chloroplast cup, convergence zones at the rim of the cup are located far from the site of thylakoid biogenesis and only contain mature thylakoids. One possibility is that the rim could be the site where old and photodamaged thylakoid proteins are transported out of the chloroplast for degradation. In this source-sink model, there would be a constant flux of proteins from the source at the chloroplast base to the sink at the rim of the cup. It is also worth considering that the apical rim of the chloroplast is positioned close to the flagella, the only part of the cell's external membrane that comes into direct contact with the extracellular environment (**Figure 1A**). The rim is thus well situated for the uptake and release of molecules that cannot permeate the cell wall. It is possible that thylakoid tip convergence zones are a conserved feature of algal chloroplasts, as similar structures can be found in a wide range of algae (**Gibbs, 1962**; **Moisan et al., 2006**). Furthermore, thylakoid tip convergence zones may be homologous to the biogenesis centers of cyanobacteria (**Kunkel, 1982**; **van de Meene et al., 2006**; **Stengel et al., 2012**).

## Pyrenoid tubules bridge the spatially separated light-dependent and light-independent steps of photosynthesis

Although the aggregation of RuBisCO within the pyrenoid is advantageous for utilizing the high local concentration of $CO_2$, this causes RuBisCO to be physically isolated from the photosynthetic pathways that occur in the thylakoid stacks and chloroplast stroma. The pyrenoid tubules appear to solve this problem, as they connect these regions of the chloroplast by serving as conduits through the dense starch sheath.

The RuBisCO catalytic reaction does not use ATP or NADPH. However, RuBisCO activase is also localized to the pyrenoid (**McKay and Gibbs, 1991**) and requires ATP to promote RuBisCO activity (**Streusand and Portis, 1987**). While immunolabeling data indicates that many Calvin–Benson cycle enzymes may be restricted to the stroma (**Suss et al., 1995**), phosphoribulokinase (PRK), which uses ATP to carry out the final step of ribulose-1,5-bisphosphate (RuBP) regeneration, has been detected in close association with the pyrenoid (**Holdsworth, 1971**; **McKay and Gibbs, 1991**). Thus, the pyrenoid likely requires ATP from the chloroplast stroma. Conversely, the product of carbon fixation, G3P, must be exported from the pyrenoid. If other Calvin–Benson cycle enzymes do indeed exclusively reside in the stroma, then regeneration intermediates of the Calvin–Benson cycle would also need to exchange efficiently between the stroma and the pyrenoid.

The pyrenoid minitubules identified in this study (**Figure 9E–H**) form narrow continuous channels between the interthylakoid stromal space and the pyrenoid matrix (**Figures 10 and 11**). Therefore, we propose that minitubules facilitate the rapid targeted diffusion of small molecules such as ATP and

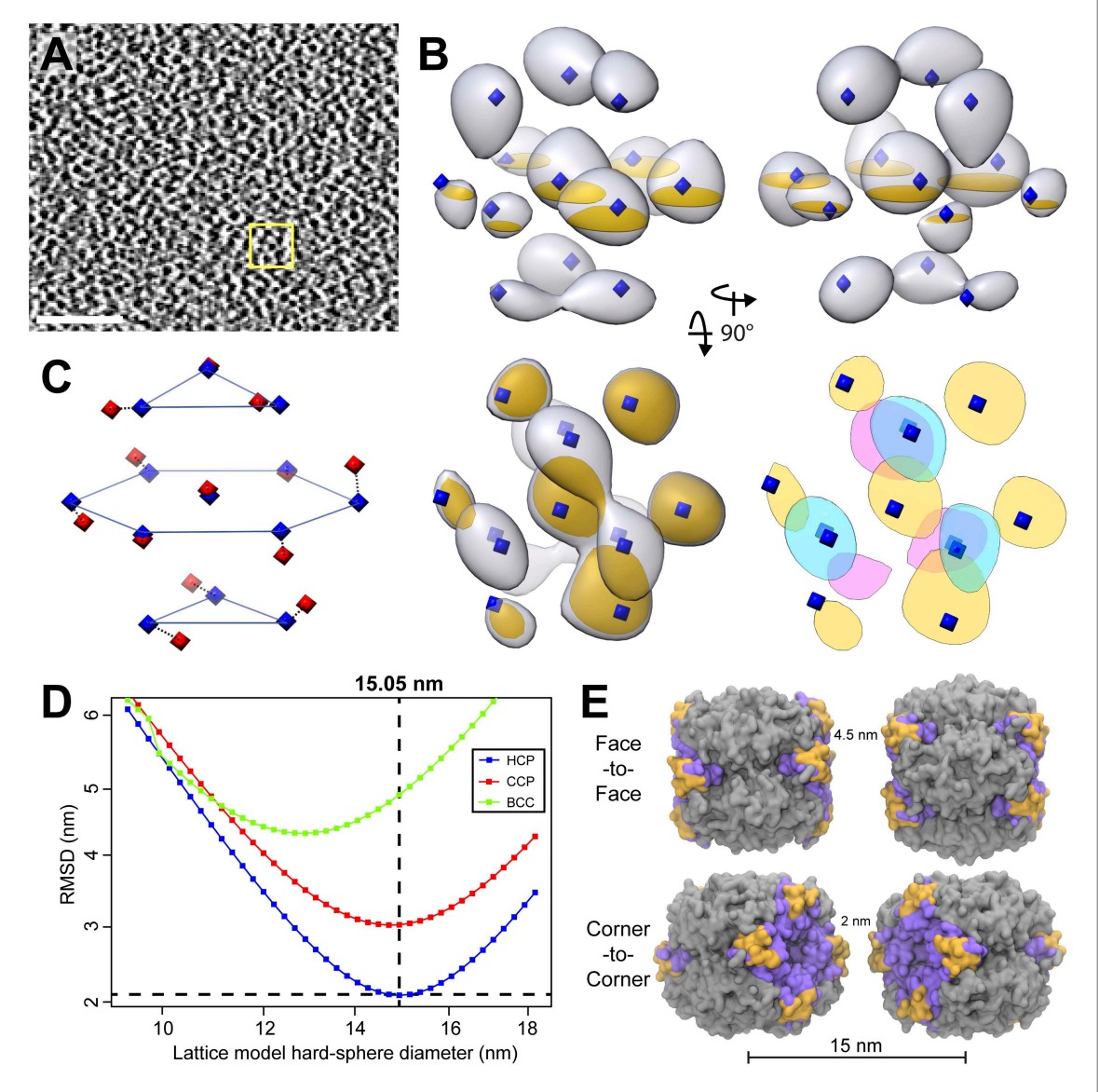

**Figure 12**. RuBisCO complexes in the pyrenoid show hexagonal close packing. (**A**) A slice from a tomographic volume containing pyrenoid RuBisCO complexes. The yellow box indicates the size of the 45.6 nm³ subvolumes used for averaging. Unbinned pixel size: 5.7 Å. Scale bar: 100 nm. (**B**) Alignment and averaging of 10,625 subvolumes without imposing symmetry yielded a hexagonally packed 3D average (grey densities, 34.8 Å resolution at the FSC 0.5 cutoff). A 2D slice through this average (gold discs) is displayed to help illustrate the hexagonal arrangement. This average fits well to the ideal center points of hexagonal close packing (blue diamonds). In the lower right panel, a top view of 2D slices through the bottom (red), middle (yellow), and top (blue) layers of the average clearly shows the A-B-A arrangement of RuBisCO complexes. (**C**) The best fit between the ideal center points of hexagonal close packing (blue diamonds) and the center points of RuBisCO nearest neighbors determined by k-means clustering a 3D point cloud of template-matched RuBisCO positions (**Figure 13**) (red diamonds). (**D**) RMSD minimization of hard-sphere models of hexagonal close packing (HCP), cubic close packing (CCP), and body-centered cubic (BCC) packing with the nearest neighbor RuBisCO positions depicted in **C**. Varying the sphere diameter yields a unique minimum RMSD for each model. HCP with a sphere diameter of 15.05 nm shows the best fit, with an RMSD of 2.13 nm. (**E**) Two possible relative orientations of neighboring RuBisCO complexes with 15 nm between their centers. Grey: RuBisCO large subunit, purple: RuBisCO small subunit, orange: the two surface-exposed α-helices on the small subunit with hydrophobic residues required for pyrenoid formation. When the sides of two RuBisCO complexes directly face each other (face-to-face), there is ~4.5 nm between the small subunits of each complex. If each complex is rotated 45° from the face-to-face orientation, opposing small complexes are ~2 nm apart (corner-to-corner). This 2 nm separation is the smallest possible distance between two complexes. Model generated from the *Chlamydomonas* RuBisCO crystal structure (*Taylor et al., 2001*).

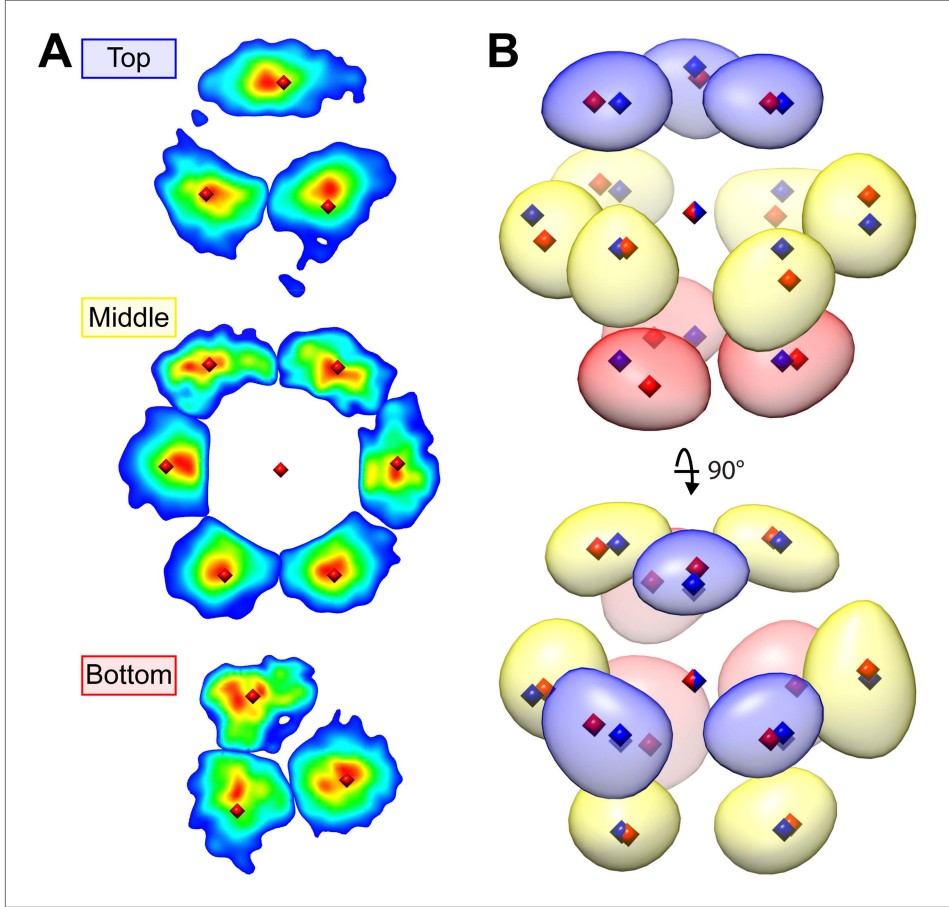

**Figure 13.** Hexagonal close packing of RuBisCO complexes determined by template matching and nearest neighbor point cloud clustering. Each RuBisCO complex was localized in the tomograms by template matching. The nearest neighbors of each RuBisCO were overlaid into a 3D point cloud by applying the transformations obtained from subvolume alignment. (**A**) 2D slices through the top, middle, and bottom of the point cloud. Color indicates point density (red: highest density, blue: lowest density). Center positions for each of the nearest neighbors (red diamonds) were determined by applying k-means clustering to the point cloud. (**B**) A Gaussian-smoothed representation of the 3D point cloud showing the clear A-B-A arrangement of hexagonal close packing. The blue, yellow, and red densities correspond to the top, middle, and bottom slices in **A**. Red diamonds: center positions generated by k-means clustering, blue diamonds: ideal center points of hexagonal close packing. DOI: 10.7554/eLife.04889.022

Calvin–Benson cycle sugars between these two compartments. The minitubule lumen diameter is only 3–4 nm by 8–15 nm, while the longest axes of ATP and the sugars RuBP and G3P measure ~1.4 nm, ~1.2 nm, and ~0.7 nm, respectively. Thus, diffusion of these metabolites through the minitubules would be practically one-dimensional. While we are unsure how the local minitubule environment may affect the diffusion coefficient, assuming a diffusion coefficient of a small molecule in the cytoplasm (*Milo et al., 2010*), it would only take a few milliseconds to travel 1 μm through a minitubule.

In addition to bundling minitubules and directing them towards the pyrenoid center, the larger pyrenoid tubules may play an important role in supplying RuBisCO with saturating levels of $CO_2$. It is believed that $HCO_3^-$ in the thylakoid lumen is converted to $CO_2$ by carbonic anhydrase 3 (CAH3) (*Karlsson et al., 1998*; *Hanson et al., 2003*), a reaction that is aided by the high $H^+$ concentration in the lumen (*Raven, 1997*). Since membranes are much more permeable to $CO_2$ than $HCO_3^-$ (*Moroney et al., 2011*), $CO_2$ would then freely diffuse out of the thylakoids and into the pyrenoid. Under low-$CO_2$ conditions, CAH3 is phosphorylated and moves from the thylakoids to the pyrenoid tubules, where it can efficiently supply RuBisCO with $CO_2$ (*Blanco-Rivero et al., 2012*; *Sinetova et al., 2012*). Thus, while minitubules may deliver the energy and sugars for carbon fixation, the larger pyrenoid tubules likely provide the $CO_2$.

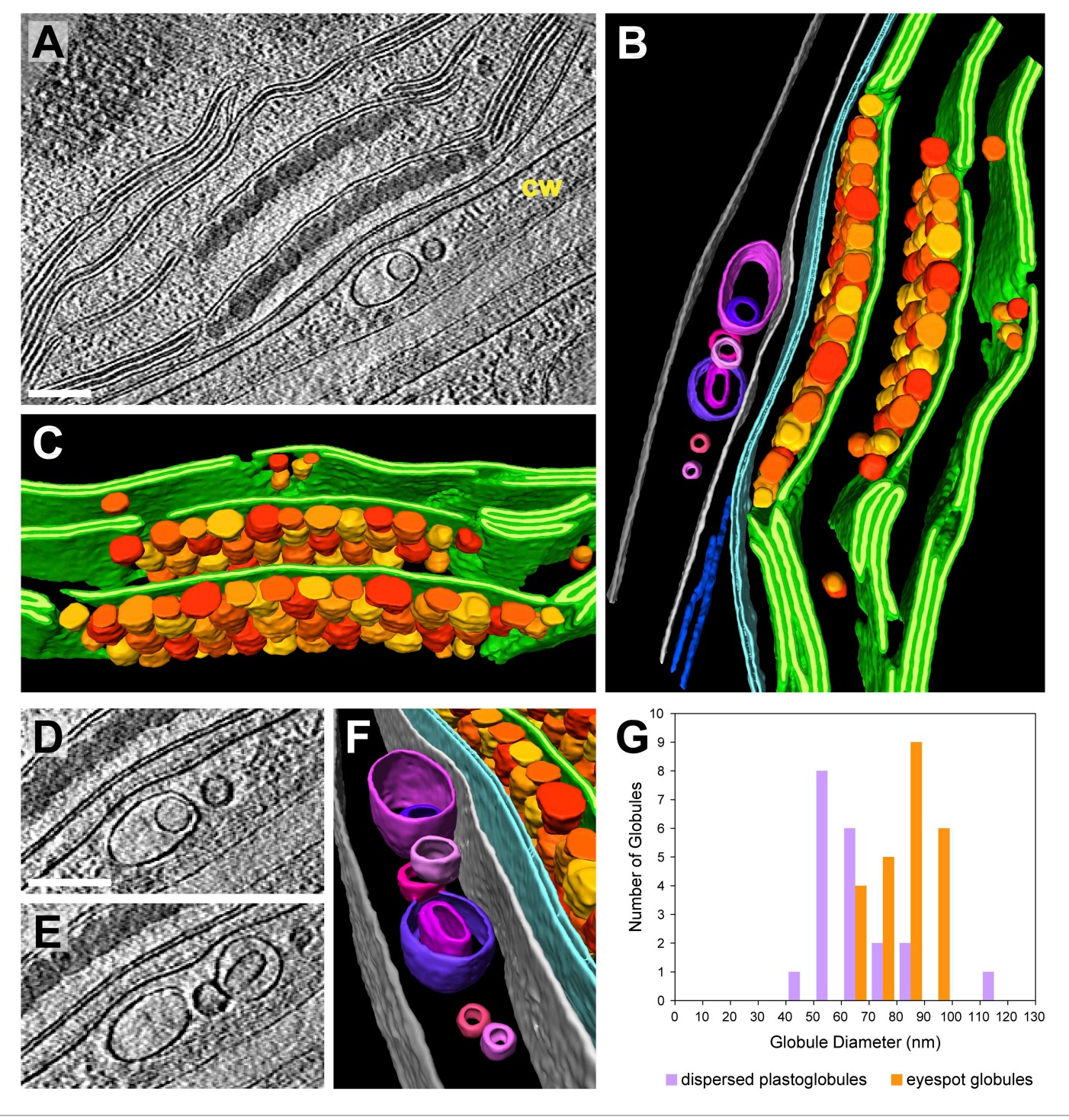

**Figure 14**. 3D architecture of the eyespot. (**A**) A slice from a tomographic volume (cw: cell wall) and (**B**) corresponding 3D segmentation (rotated ~90° from **A**) showing thylakoid membranes (dark green), thylakoid lumina (light green), eyespot globules (yellow, orange, and red), the chloroplast envelope (light blue), flagellar root microtubules (dark blue), the plasma membrane (light grey), the outer border of the cell wall (dark grey), and cell wall vesicles (purple and pink). (**C**) A view of the segmentation with the chloroplast envelope and plasma membrane removed, revealing the close-packed arrangement of eyespot globules along specialized thylakoid regions. (**D**–**E**) Two sequential tomogram slices and (**F**) corresponding 3D segmentation showing vesicular bodies adjacent to the eyespot, embedded between the plasma membrane and the outer border of the cell wall. (**G**) Histogram comparing eyespot globule sizes with the sizes of plastoglobules found elsewhere in the chloroplast. The tomogram was 2× binned. Unbinned pixel size: 9.6 Å. Segmented tomogram thickness: 210 nm. Scale bars: 200 nm.

The structure of the pyrenoid is dynamic and responds to environmental changes including light, temperature, and atmospheric composition. When cells are grown in low light, the pyrenoid matrix becomes fenestrated with more pyrenoid tubules (*Rawat et al., 1996*). Such an inducible condition may allow the examination of pyrenoid tubule assembly and growth through the pyrenoid matrix.

## Hexagonal close packing of RuBisCO: specific inter-complex interactions or random packing caused by pyrenoid forces?

The hexagonal arrangement of RuBisCO complexes within the pyrenoid could be caused by one of two distinct influences. Either there are specific interactions between RuBisCO complexes that enforce this hexagonal configuration, or non-specific packing forces resulting from crowding within the pyrenoid compress the complexes together, yielding hexagonal close packing of roughly spherical objects. If RuBisCO complexes do make specific interactions with each other, this would indicate that there may be allosteric molecular mechanisms for increasing the efficiency of carbon fixation. Two studies found evidence that the RuBisCO small subunit may mediate inter-complex interactions in *Chlamydomonas* (*Genkov et al., 2010*; *Meyer et al., 2012*). The hydrophobic residues in the surface-exposed α-helixes of the RuBisCO small subunit were shown to be necessary for pyrenoid formation, and RuBisCO aggregation was prevented in vivo when these domains were replaced by less hydrophobic sequences from higher plants. This implies that hydrophobic interactions between RuBisCO complexes may drive pyrenoid formation.

Alternatively, the hexagonal arrangement of RuBisCO particles may be caused by non-specific pyrenoid packing forces. We found that the centers of RuBisCO complexes in the pyrenoid were ~15 nm apart (*Figure 12D*), even though RuBisCO is shaped like a rounded cube with a shortest diameter of ~10.5 nm and a longest diameter of ~13 nm. Thus, there is a 2–4.5 nm space between RuBisCO complexes, depending on their relative orientations (*Figure 12E*). This data suggests that RuBisCO complexes within the pyrenoid may not directly interact with each other and that other factors, such as a dense matrix of proteins and carbohydrates or small linker proteins between the complexes (*Meyer and Griffiths, 2013*), determine the RuBisCO spacing. In the case of non-specific packing, the key question becomes how these packing forces are generated. One possibility is that since the pyrenoid is encased in a dense starch sheath, packing forces could increase as additional RuBisCO complexes are inserted into this relatively fixed volume. Such a model raises the question of how RuBisCO complexes are targeted through the starch sheath during pyrenoid growth.

In different classes of bacterial carboxysomes, both random and specific packing of RuBisCO have been observed (*Rae et al., 2013*). In α-carboxysomes, which are found primarily in ocean-dwelling cyanobacteria, non-specific packing forces alone are believed to be sufficient to induce the layered organization of RuBisCO (*Iancu et al., 2010*). However, in the β-carboxysomes of freshwater cyanobacteria, RuBisCO complexes are held in an organized array by CcmM linker proteins (*Long et al., 2007*, *2010*; *Peña et al., 2010*).

While our data cannot discriminate between random and specific causes of hexagonal RuBisCO packing in the *Chlamydomonas* pyrenoid, several lines of investigation may be able to help tease apart this question. Technical advances such as direct detection cameras continue to improve the resolution of cryo-tomograms. Once higher resolutions are attained, it will be possible to use template matching to determine whether RuBisCO complexes have a preferred orientation relative to each other, a strategy that has already been successfully applied to ribosomes (*Brandt et al., 2009*, *2010*; *Pfeffer et al., 2012*). If RuBisCO complexes within the pyrenoid do indeed have preferred orientations, template matching will also reveal which surfaces likely mediate RuBisCO aggregation. A complementary approach would be to examine how RuBisCO packing changes under conditions where the pyrenoid becomes less organized, including mutant strains (*Thyssen et al., 2003*; *Ma et al., 2011*), elevated levels of atmospheric $CO_2$ (*Rawat et al., 1996*; *Borkhsenious et al., 1998*), and during RuBisCO aggregation in newly divided cells at dawn (*Mitchell et al., 2014*). If the hexagonal packing disappears in disorganized pyrenoids, this would indicate that the packing is driven by non-specific pyrenoid forces. However, if islands of hexagonal packing persist even when RuBisCO complexes are not being forced together, this would provide evidence for inter-complex interactions. Additionally, the pyrenoid forms in mutants that lack a starch sheath (*Villarejo et al., 1996*), enabling a direct test of whether the starch induces hexagonal packing of RuBisCO.

By combining cryo-FIB with cryo-ET, we acquired the first in situ 3D views of the algal chloroplast, native structures that provide new insights into the mechanisms of thylakoid biogenesis,

photosynthesis, and carbon fixation. The challenge now lies in characterizing the identity, localization, and geometry of the macromolecular complexes that orchestrate the chloroplast's elegant architecture.

# Materials and methods

## Cell culture

*C. reinhardtii* wild-type strain CC-124 (137c) was acquired from the *Chlamydomonas* Resource Center (University of Minnesota, Minneapolis, MN) and grown in Tris-acetate-phosphate (TAP) medium (*Harris et al., 2009*) with constant light (~10,000 lx) and aeration with normal atmosphere. CC-124 was chosen over the CC-125 wild-type strain due to smaller cell size, which yields better vitrification by plunge-freezing.

## Plunge-freezing vitrification

5 µl of liquid culture (diluted to 1000–750 cells/µl in fresh TAP) was pipetted onto holey carbon-coated 200 mesh copper grids (Quantifoil Micro Tools, Jena, Germany). The grids were blotted from the reverse side and immediately plunged into a liquid ethane/propane mixture at liquid nitrogen temperature using a Vitrobot Mark 4 (FEI, Eindhoven, The Netherlands). The Vitrobot was set to 25°C, 90% humidity, blot force 10, and 4–7 s blot time. The grids were stored in sealed boxes in liquid nitrogen until used.

## Cryo-FIB milling

Cryo-FIB milling (*Rigort et al., 2012b*) was performed as described in detail at Bio-protocol (*Schaffer et al., 2015*). Plunge-frozen grids were mounted into autogrids that were modified for FIB milling (Autogrid sample holder, FEI), providing stability to the EM grids during sample preparation and transfers. These autogrids were mounted into a dual-beam (FIB/SEM) microscope (Quanta 3D FEG, FEI) using a cryo-transfer system (PP3000T, Quorum) with a custom-built transfer shuttle. During FIB operation, samples were kept at a constant temperature of −180°C using a homemade 360° rotatable cryostage (*Rigort et al., 2010*, *2012a*). To improve sample conductivity and reduce curtaining artifacts during FIB milling, the whole Autogrid was coated by organometallic platinum (protocol adapted from *Hayles et al., 2007*) using the in situ gas injection system (GIS, FEI) and the following parameters: 13.5 mm stage working distance, 26°C GIS pre-heating temperature, and 4 s gas injection time. Thin lamellas were prepared using the Ga+ ion beam at 30 kV under a shallow 8°–12° angle. Rough milling was performed with a rectangular pattern and 0.3 nA beam current, followed by sequentially lowered beam currents of 0.1 nA, 50 pA and 30 pA during the thinning and cleaning steps. During the milling, the specimen was also imaged using the SEM at 3–5 kV and 5–30 pA.

## Cryo-ET

Tomography was performed on a Tecnai G2 Polara transmission electron microscope (FEI, Eindhoven, Netherlands) equipped with a field emission gun operated at 300 kV, a GIF 2002 post-column energy filter (Gatan, Pleasanton, CA), and either a 2048 × 2048 Gatan slow scan CCD camera or a 3838 × 3710 Gatan K2 Summit direct detection camera (only *Figures 7 and 11A–E*). Tilt-series acquisition was controlled by SerialEM software (*Mastronarde, 2005*) under low-dose conditions (~100 e/Å$^2$ cumulative dose). CCD images were recorded at 2° tilt increments, with −8 µm to −12 µm defocus, at 31,500×, 42,000×, and 52,500× magnifications (pixel sizes of 9.6 Å, 7.1 Å, and 5.7 Å). 52,500× was the highest possible magnification that still prevented excessive electron damage to the sample. K2 images were recorded at 2° tilt increments, with −5 µm defocus, at 11,800× and 14,500× magnifications (pixel sizes of 4.2 Å and 3.4 Å). The dataset for this study consisted of 43 tomograms from 27 FIB lamella preparations.

## Tomogram reconstruction and segmentation

Fiducial-less alignment of the tilt-series was performed by feature tracking with TOM software (*Nickell et al., 2005*; *Korinek et al., 2011*) or patch tracking with IMOD software (*Mastronarde, 1997*), and tomograms were reconstructed using the weighted back projection algorithm. Segmentation of tomogram volumes was performed with Amira software (FEI Visualization Sciences Group).

## Template matching and subtomogram averaging

Subtomograms of pyrenoid regions were binned once to a pixel size of 11.4 Å for localization of RuBisCO complexes. RuBisCO complexes were detected by template matching (*Frangakis et al., 2002*)

using a spherical template with a diameter of 13.68 nm. A spherical template, which functioned well due to the homogeneity of pyrenoid composition and the roughly spherical shape of RuBisCO at the tomogram resolution, made it possible to discard the rotational search, dramatically increasing the localization speed and enabling iterative fine-tuning of parameter values. For template matching, the pyrenoid subtomograms were low-pass filtered using the Crowther criterion (*Crowther et al., 1970*) to the resolution of a 12 nm diameter particle with the same 2° tilt increment as the tomographic tilt series. Cross-correlation peaks were exhaustively extracted, yielding a set of RuBisCO particle positions. Over 10,000 subvolumes (45.6 nm$^3$) centered on these RuBisCO positions were selected from the unfiltered tomograms in order to analyze the local neighborhood around each RuBisCO complex. The subvolumes were subsequently aligned and averaged with the PyTom toolbox as previously described (*Hrabe et al., 2012*; *Chen et al., 2013*).

## Nearest neighbor analysis of RuBisCO organization

For each RuBisCO position detected by template matching, a range query (*Bentley, 1975*) was performed to select a neighborhood of RuBisCO positions within a 22.8 nm radius. The transformations obtained by subvolume alignment were applied to their corresponding point neighborhoods to generate a cloud of 3D points describing the local arrangement of RuBisCO complexes in the same reference frame as the subvolume average. K-means clustering of the 3D point cloud (with k = 13) yielded the center positions of the reference RuBisCO and its 12 nearest neighbors. Subsequently, the unit cells of different hard sphere lattice arrangements (hexagonal close packing, cubic close packing, and body-centered cubic) were simulated for a series of sphere diameters. The center points of these hard spheres were compared to the cluster center positions by computing the RMSD values of nearest point correspondences, revealing the packing model and sphere diameter that best matched the data.

## Molecular measurements and modeling

Atomic models of RuBisCO complexes with different relative orientations were generated with Visual Molecular Dynamics (VMD) (*Humphrey et al., 1996*). Molecular dimensions and inter-complex distances were measured with VMD and UCSF Chimera (*Pettersen et al., 2004*).

## Acknowledgements

We are grateful to Yoshiyuki Fukuda for assistance with Vitrobot sample preparation, Tim Laugks for help with *Figure 1D*, Friedrich Förster, Georg Kreimer, Stefan Geimer, James Moroney, and Wallace Marshall for advice, and Karin Engel for critical reading of the manuscript. This work was supported by the European Commission's 7th Framework Programme grant agreement HEALTH-F4-2008-201648/PROSPECTS, a postdoctoral research fellowship from the Alexander von Humboldt Foundation (to BDE), a CONACyT-DAAD graduate scholarship (to LKC), the Deutsche Forschungsgemeinschaft Excellence Clusters CIPSM and SFB 1035 (to WB), the Federal Ministry of Education and Research (BMBF), and an inter-institutional research initiative of the Max Planck Society.

## Additional information

### Funding

| Funder | Grant reference number | Author |
| --- | --- | --- |
| European Commission | 7th Framework Programme Grant HEALTH-F4-2008-201648/PROSPECTS | Wolfgang Baumeister |
| Alexander von Humboldt-Stiftung (Humboldt Foundation) | Postdoctoral Research Fellowship | Benjamin D Engel |
| German Academic Exchange Service | CONACyT-DAAD Graduate Scholarship | Luis Kuhn Cuellar |
| Deutsche Forschungsgemeinschaft | Excellence Clusters CIPSM and SFB 1035 | Wolfgang Baumeister |
| Bundesministerium für Bildung und Forschung (Federal Ministry of Education and Research) | | Wolfgang Baumeister |

| Funder | Grant reference number | Author |
|---|---|---|
| Max-Planck-Gesellschaft (Max Planck Society) | Inter-Institutional Research Initiative | Wolfgang Baumeister |

The funders had no role in study design, data collection and interpretation, or the decision to submit the work for publication.

## Author contributions

BDE, Conception and design, Acquisition of data, Analysis and interpretation of data, Drafting or revising the article; MS, Acquisition of data, Analysis and interpretation of data, Drafting or revising the article; LKC, EV, Analysis and interpretation of data, Drafting or revising the article; JMP, Conception and design, Acquisition of data, Drafting or revising the article; WB, Conception and design, Drafting or revising the article

## Author ORCIDs

Benjamin D Engel, http://orcid.org/0000-0002-0941-4387

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
