## [Decision Letter]

Thank you for sending your work entitled “Native Architecture of the *Chlamydomonas* Chloroplast Revealed by In Situ Cryo-Electron Tomography” for consideration at *eLife*. Your article has been favorably evaluated by Vivek Malhotra (Senior editor), a Reviewing editor, and 2 reviewers.

Joanne Chory (Reviewing editor) and Howard Griffiths (peer reviewer) have agreed to reveal their identity.

The Reviewing editor and the reviewers discussed their comments before we reached this decision, and the Reviewing editor has assembled the following comments to help you prepare a revised submission.

This paper uses the techniques of focused ion beam milling and cryo-electron tomography to define inner compartments of *Chlamydomonas* chloroplasts at a level of detail unimaginable even a few years ago. Although the results are somewhat descriptive, the reviewers agree that the degree of resolution is remarkable and allows the authors to address chloroplast biology and the C-concentrating mechanism. The reviewers feel that this paper has the potential to become a landmark paper, and request that revisions be made to the Discussion and to a couple of figures.

1) Colour coding for some of the graphics could be improved. For instance, Figure 8 c.f. 8F, H; these are compelling images, revealing the internal structure of the intra-pyrenoidal “knotted” thylakoid membranes to an amazing level of detail. But the colour coding is confusing: in 8B, the reconstruction reveals that the large, merged external thylakoid membrane contains (what become apparent as) green mini-tubules (8F, H); however, in 8F, H, this colour coding for the internal minitubules is changed to yellow, which has previously been used to indicate intrathylakoid lumen. If both were green some of the contrast might be lost with the outer, encasing thylakoid membrane, but the coding would be consistent. Any change made should also be adopted in Figure 10, where again we see the minitubules emerging into the heart of the pyrenoid.

2) Changes to the Discussion section: are there differences or artifacts in previously published work that come into question? The RuBisCO results may be over-emphasized, which may mislead some readers. It is most likely that the appearance of hexagonal packing is the result of randomly arranging RuBisCO within a confined space. This should be made clear in the text.

---

## [Author Response]

*1) Colour coding for some of the graphics could be improved. For instance,*
Figure 8
*c.f. 8F, H; these are compelling images, revealing the internal structure of the intra-pyrenoidal “knotted” thylakoid membranes to an amazing level of detail. But the colour coding is confusing: in 8B, the reconstruction reveals that the large, merged external thylakoid membrane contains (what become apparent as) green mini-tubules (8F, H); however, in 8F, H, this colour coding for the internal minitubules is changed to yellow, which has previously been used to indicate intrathylakoid lumen. If both were green some of the contrast might be lost with the outer, encasing thylakoid membrane, but the coding would be consistent. Any change made should also be adopted in*
Figure 10*, where again we see the minitubules emerging into the heart of the pyrenoid*.

We agree it is confusing that the light green thylakoid lumina and the yellow minitubules are similar in color. Thus, we have changed the minitubules to orange and updated Figures 9 and 11 and Videos 4 and 5 accordingly.

2) Changes to the Discussion section: are there differences or artifacts in previously published work that come into question?

Our approach of combining cryo-FIB with cryo-ET yielded data with several key advantages over earlier studies that relied on classical plastic embedding and sectioning. Our tomograms represent the artifact-free native state of hydrated cells, reveal 3D views of the cellular environment, and provide much higher resolution information. This enabled us to make observations about cellular architecture that were not previously possible. We make this point generally in the Introduction: “Much of our understanding of chloroplast architecture comes from conventional transmission electron microscopy (TEM) studies (44; 45; 89; 90; 102) using the traditional protocol of sample fixation, heavy metal staining, dehydration, plastic embedding, and sectioning with an ultramicrotome. Each of these preparation steps can distort structures and obscure high-resolution information (123). Sample preparation by freeze-fracture has enabled observations of integral proteins embedded in membrane surfaces (32; 91), but it cannot provide a 3D view of chloroplast architecture. Vitrification by rapid freezing offers the best possible preservation of biological material (22). Imaging vitreous samples by cryo-electron tomography (cryo-ET) generates 3D views of native structures, where image contrast corresponds to the intrinsic variation in mass density instead of the sample’s local affinity for heavy metal stains (7).”

Specifically, our technique enabled the first detailed analysis of fine and transient features including: 1) chloroplast envelope inner membrane invaginations, 2) pyrenoid minitubules, and 3) the packing of RuBisCO complexes within the pyrenoid. While not singling out individual studies, in the Introduction we emphasize the novelty of these observations compared to all previous work: “Because all biological material was immobilized at the moment of whole cell vitrification, fine structures remained well preserved within their cellular context, enabling accurate measurements of thylakoid architecture, membrane invaginations, pyrenoid tubules, the eyespot, and the 3D organization of RuBisCO complexes within the pyrenoid matrix.”

Differences from previous classical EM studies are also noted in the manuscript:

“Pyrenoid tubules have previously been reported to contain “ridges” or “infoldings” along the inner surface of the tubule membrane (89). Interestingly, in high magnification cryo-tomograms these structures were revealed to be smaller tubules (Figure 9 and Figure 9—figure supplement 1), which we have termed pyrenoid minitubules.”

“The pyrenoids of most species, including those found in *Chlamydomonas*, appear amorphous (36; 80). Since these conclusions about pyrenoid architecture are based on 2D EM of plastic sections, we decided to investigate whether RuBisCO complexes were ordered within our native-state tomographic volumes of the *Chlamydomonas* pyrenoid. (…) symmetry-free subtomogram averaging of 45.6 nm^3^ subvolumes revealed that the RuBisCO complexes were actually hexagonally packed (Figure 12).”

*The RuBisCO results may be over-emphasized, which may mislead some readers. It is most likely that the appearance of hexagonal packing is the result of randomly arranging RuBisCO within a confined space. This should be made clear in the text*.

Given the available studies on RuBisCO packing, it is premature to make the judgment that “it is most likely that the appearance of hexagonal packing is the result of randomly arranging RuBisCO within a confined space”. Both random packing and specific packing interactions are plausible mechanisms. In our discussion, we discuss the evidence in favor of each mechanism without showing favoritism towards either model. The evidence in favor of specific packing in *Chlamydomonas* is outlined in the Discussion section: “Two studies found evidence that the RuBisCO small subunit may mediate inter-complex interactions in *Chlamydomonas* (30; 80). The hydrophobic residues in the surface-exposed α-helixes of the RuBisCO small subunit were shown to be necessary for pyrenoid formation, and RuBisCO aggregation was prevented in vivo when these domains were replaced by less hydrophobic sequences from higher plants. This implies that hydrophobic interactions between RuBisCO complexes may drive pyrenoid formation.”

For comparison, we have added a new paragraph to the Discussion explaining how different types of carboxysomes can employ either specific or random packing mechanisms: “In different classes of bacterial carboxysomes, both random and specific packing of RuBisCO have been observed (96). In α-carboxysomes, which are found primarily in ocean-dwelling cyanobacteria, non-specific packing forces alone are believed to be sufficient to induce the layered organization of RuBisCO (53). However, in the β-carboxysomes of freshwater cyanobacteria, RuBisCO complexes are held in an organized array by CcmM linker proteins (72; 73; 92).”

As stated in the Discussion, our data favors either random packing or linker proteins, not direct inter-RuBisCO interactions: “Alternatively, the hexagonal arrangement of RuBisCO particles may be caused by non-specific pyrenoid packing forces. We found that the centers of RuBisCO complexes in the pyrenoid were ∼15 nm apart (Figure 12), even though RuBisCO is shaped like a rounded cube with a shortest diameter of ∼10.5 nm and a longest diameter of ∼13 nm. Thus, there is a 2-4.5 nm space between RuBisCO complexes, depending on their relative orientations (Figure 12). This data suggests that RuBisCO complexes within the pyrenoid may not directly interact with each other and that other factors, such as a dense matrix of proteins and carbohydrates or small linker proteins between the complexes (79), determine the RuBisCO spacing.”

To further emphasize that non-specific packing forces result in “random” packing of RuBisCO, we have modified the title of the section “Hexagonal close packing of RuBisCO: specific inter-complex interactions or *random* packing caused by pyrenoid forces?”, as well as the following sentence: “While our data cannot definitively judge between random and specific causes of hexagonal RuBisCO packing in the *Chlamydomonas* pyrenoid, several lines of investigation may be able to help tease this question apart.”